# Distinct spermiogenic phenotypes underlie sperm elimination in the *Segregation Distorter* meiotic drive system

**Marion Herbette**[1], **Xiaolu Wei**[2], **Ching-Ho Chang**[3¤], **Amanda M. Larracuente**[3], **Benjamin Loppin**[1], **Raphaëlle Dubruille**[1]*

**1** Laboratoire de Biologie et Modélisation de la Cellule, CNRS UMR 5239, École Normale Supérieure de Lyon, University of Lyon, Lyon, France, **2** University of Rochester Medical Center, Department of Biomedical Genetics, Rochester, New York, United States of America, **3** University of Rochester Department of Biology, Rochester, New York, United States of America

¤ Current address: Fred Hutchison Cancer Research Center, Seattle, Washington, United States of America
* raphaelle.renard-dubruille@ens-lyon.fr

## Abstract

*Segregation Distorter* (*SD*) is a male meiotic drive system in *Drosophila melanogaster*. Males heterozygous for a selfish *SD* chromosome rarely transmit the homologous *SD⁺* chromosome. It is well established that distortion results from an interaction between *Sd*, the primary distorting locus on the *SD* chromosome and its target, a satellite DNA called *Rsp*, on the *SD⁺* chromosome. However, the molecular and cellular mechanisms leading to post-meiotic *SD⁺* sperm elimination remain unclear. Here we show that *SD/SD⁺* males of different genotypes but with similarly strong degrees of distortion have distinct spermiogenic phenotypes. In some genotypes, *SD⁺* spermatids fail to fully incorporate protamines after the removal of histones, and degenerate during the individualization stage of spermiogenesis. In contrast, in other *SD/SD⁺* genotypes, protamine incorporation appears less disturbed, yet spermatid nuclei are abnormally compacted, and mature sperm nuclei are eventually released in the seminal vesicle. Our analyses of different *SD⁺* chromosomes suggest that the severity of the spermiogenic defects associates with the copy number of the *Rsp* satellite. We propose that when *Rsp* copy number is very high (> 2000), spermatid nuclear compaction defects reach a threshold that triggers a checkpoint controlling sperm chromatin quality to eliminate abnormal spermatids during individualization.

## Author summary

In diploid organisms, both alleles of the same gene have an equal chance to be transmitted to the progeny. However, in many species including mammals, insects and plants, selfish genetic elements perturb gametogenesis in a way that favors their own transmission to the detriment of the homologous chromosome that does not carry them. In *Drosophila melanogaster*, *Segregation Distorter* (*Sd*) is a well-characterized selfish locus that induces, by still-unclear mechanisms, the elimination of sperm cells which contain the homologous second chromosome when this carries a large heterochromatic block of repetitive DNA

**Data Availability Statement:** All relevant data are within the manuscript and its Supporting Information files.

**Funding:** This work was supported by a grant to RD from the Agence Nationale de La Recherche (ANR-16-CE12-0006-01) and a grant to AML from the National Institutes of Health (R35 GM119515). The funders had no role in study design, data collection and analysis, decision to publish, or preparation of the manuscript.

**Competing interests:** The authors have declared that no competing interests exist.

called *Rsp*. Here, we show that in *Sd* males, the replacement of histones by sperm specific protamine-like proteins is perturbed in the differentiating *Rsp* sperm cells, which are then eliminated before their release in the seminal vesicle. However, in some genetic backgrounds, many spermatids *Rsp* tend to escape this elimination and defective sperm nuclei are found in the seminal vesicle. We show that these phenotypes are partly linked to the number of repeats in the *Rsp* block and can be modulated by suppressors present in the genetic background. Our work thus helps to understand how selfish loci exploit gametogenesis to favor their own transmission and highlights the essential role of heterochromatin in spermiogenesis progression.

## Introduction

In sexually reproducing organisms, the production of haploid gametes from diploid germ cells ensures that two alleles of the same locus are equally transmitted to the progeny. However, selfish genetic elements recurrently emerge in genomes and manipulate gametogenesis in either sex to promote their own transmission thus resulting in the distortion of Mendelian ratios [1]. Although "meiotic drivers" are widespread across plants, animals and fungi, male-specific meiotic drive systems are particularly well studied in *Drosophila* species, where 19 independent distorters are currently known [1,2]. While most of these drive systems are sex-linked and thus distort sex ratios, one of the most famous male-specific meiotic drivers is an autosomal selfish gene complex called *Segregation Distorter* (*SD*) in *D. melanogaster* [3,4].

Based on the cytological phenotype they cause, male-specific meiotic drive systems can be classified into two types: those that induce meiotic defects, such as the *Paris* sex ratio system from *D. simulans*, and those that result in post-meiotic defects—and are thus called meiotic drive systems in a broad sense [2]. *SD* is by far the best studied and documented system and belongs to the second category. It was first described in 1959 after the discovery of second chromosomes (called *SD* chromosomes) that induced distortion of the expected Mendelian ratio [3]: in the appropriate genetic background, heterozygous *SD*/*SD*$^+$ males transmit the *SD* chromosome to 95–100% of their progenies.

While *SD* systems are selfish gene complexes consisting of multiple factors that contribute to drive, two main components are molecularly characterized: *Sd*, the primary driver shared by all *SD* chromosomes, and *Responder (Rsp)*, its genetic target on *SD*$^+$ chromosome (Fig 1A). *Sd* is an incomplete duplication of the *RanGAP* gene (hereafter called *Sd-RanGAP*), that encodes a C-terminus truncated version of the Ran GTPase Activating Protein (RanGAP) [4–6]. RanGAP is a cytosolic GTPase-activating enzyme that binds to nuclear pores and hydrolyzes Ran-GTP into Ran-GDP, which aids in the transport of proteins and some RNAs from the cytosol to the nucleus [7]. The truncated Sd-RanGAP protein is still enzymatically active but mislocalizes in primary spermatocytes, the diploid cells that will give rise to spermatids after meiosis [8]. *Responder* (*Rsp*) is a satellite DNA (hereafter satDNA) that consists of dimers of two related ~120-bp AT-rich sequences that are tandemly repeated in the pericentromeric heterochromatin on the right arm of the second chromosome [9–10] (Fig 1A). The number of 120-bp *Rsp* monomers varies in natural populations and positively correlates with their sensitivity to distortion [4,9–11]. Indeed, *SD*$^+$ chromosomes with high *Rsp* copy number (e.g. >2000) are super sensitive (*Rsp*$^{ss}$) to drive and are rarely transmitted to the progeny of *SD*/*SD*$^+$ males. However, *SD*$^+$ chromosomes with intermediate *Rsp* copy number (*e.g.* 700–1000) show a continuous range of drive sensitivity (*Rsp*$^s$) and *SD*$^+$ chromosomes with low *Rsp* copy

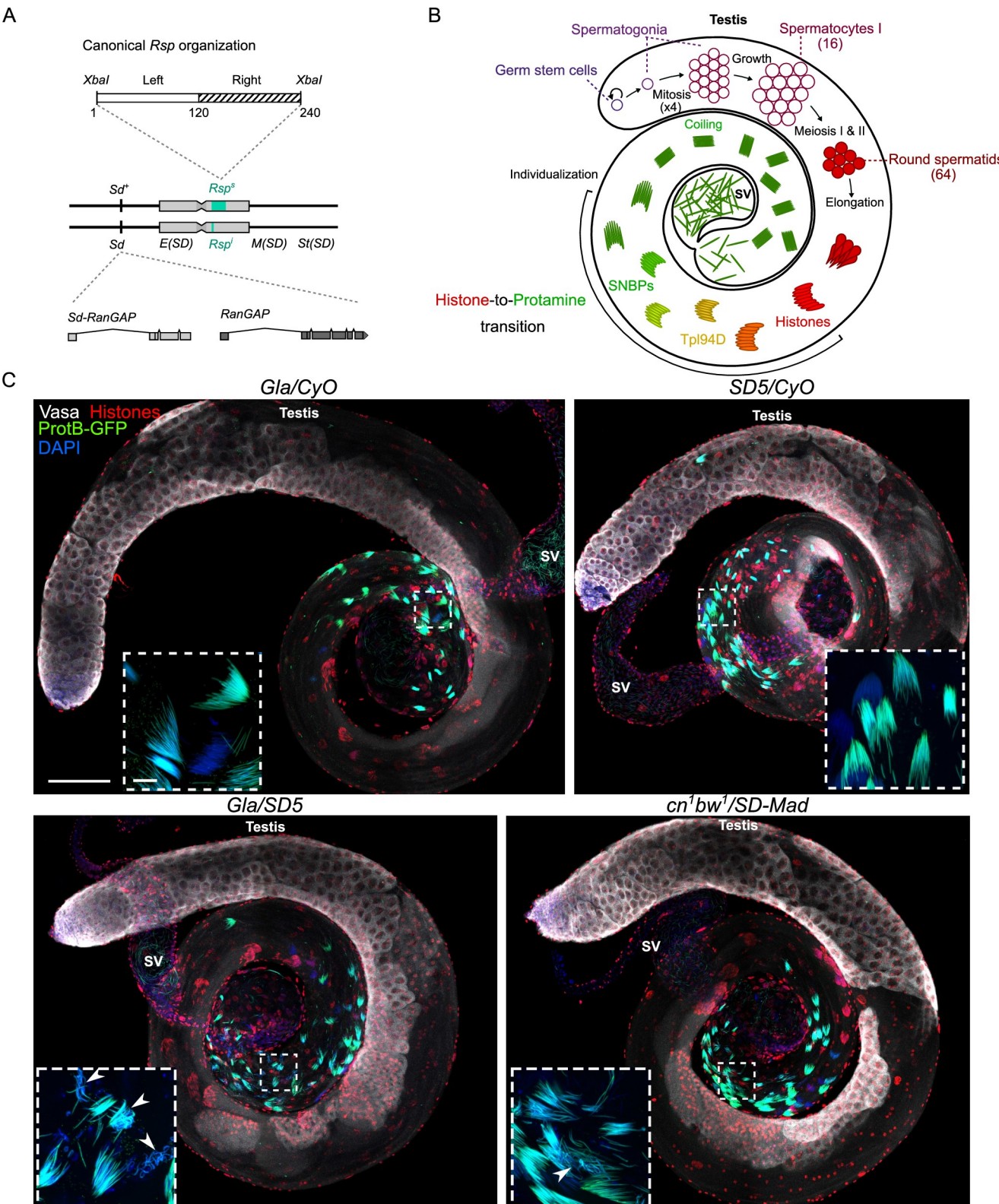

**Fig 1. In *SD* males, cytological defects are observed at the end of spermatogenesis, during the histone-to-protamine transition.** (**A**) The two main genetic elements involved in *SD* on the second chromosome. *Sd* is a duplication of the *RanGAP* gene and its target is the *Rsp* satDNA on the homologous chromosome. Gray boxes represent pericentric heterochromatin. Other uncharacterized genetic elements (*Enhancer of SD [E(SD)]; Modifier of SD [M(SD)]*

*and Stabilizer of SD [St(SD)]*) located on SD chromosomes are also required to induce high levels of distortion [4,50–52]. (**B**) A scheme of a fly testis showing the organization of spermatogenesis, which starts at the apical tip and progresses along the tubular axis of the fly testis [24]. Germ stem cells divide asymmetrically to form another germ stem cell and a spermatogonium. After four incomplete mitoses without cytokinesis, the 16 primary spermatocytes of each cyst enter meiosis and produce 64 round spermatids which are interconnected with cytoplasmic bridges and surrounded by two cyst cells (not represented on the scheme). The 64 spermatids then differentiate into mature sperm cells in synchrony. Round spermatid nuclei elongate to form needle-shaped nuclei. During this nuclear reshaping and remodeling, almost all histones (red) are removed and replaced by transition proteins (yellow) such as Tpl94D, which are then also eliminated and replaced by Mst35Ba/b, Mst77F and Prtl99 (green) during the histone-to-protamine transition. At the end of spermiogenesis, spermatids become individualized and coiled before being released in the seminal vesicle. (**C**) Confocal images of whole-mount testes from males carrying the *protB-GFP* transgene in the indicated genetic background stained with DAPI (blue) and antibodies against histones (red) and Vasa (white). In *Gla/SD5* and *cn¹ bw¹/SD-Mad* distorter males, spermatogonial amplification and meiosis appear normal and the first cytological defects are observed during the histone-to-protamine transition. Many cysts appear loosely bundled and abnormally-shaped nuclei are observed lagging behind needle-shape spermatid nuclei (insets; arrowheads). Scale bar: 100μm in large views and 10μm in magnified squares.

number (*e.g.* <200) are insensitive to drive (*Rsp^i*). *SD* chromosomes have *Rsp^i* alleles with very low *Rsp* copy number (< 20) and are thus insensitive to their own drive [4,9–11].

While identifying these key genetic elements was a major step forward in understanding the molecular principles of *SD*, the mechanisms by which these elements genetically interact and eventually lead to the specific elimination of *SD⁺ Rsp^s* gametes remain mysterious. The first obvious defects appear during spermiogenesis, the maturation of the haploid spermatids into mature sperm cells. By transmission electron microscopy, Tokuyasu and colleagues showed that about half of spermatids—supposedly the ones containing the *SD⁺* chromosome —have abnormally compacted chromatin compared to their sister spermatid nuclei [12]. Importantly, spermiogenesis is characterized by a dramatic reorganization of spermatid nuclei which reduce in volume and adopt a characteristic needle-shape [13] (see Fig 1B for a schematic description of *Drosophila* spermatogenesis). This extreme nuclear compaction is primarily driven by a global chromatin-remodeling process known as the histone-to-protamine transition, whereby most of the histones are eliminated and replaced by Sperm Nuclear Basic Proteins (SNBPs), such as the archetypal protamines in mammals [14]. In *Drosophila*, nearly all histones are eliminated at the onset of the transition and are first replaced by transition proteins, such as Tpl94D [15]. Then, transition proteins are removed and replaced by at least four protamine-like SNBPs of the MST-HMG Box family: Mst35Ba/b (ProtA/B), Prtl99C and Mst77F [16–19]. Interestingly, in 1982, Hauschteck-Jungen and Hartl showed that half of the spermatids in *SD/SD⁺* males stained weakly with a fluorescent dye that detects basic proteins [20]. The authors thus proposed that the histone-to-protamine transition—that had not yet been characterized in *Drosophila* at that time—failed to occur in the *SD⁺* nuclei. However, other studies suggested that *SD⁺* spermatids differentiated into spermatozoa and were transferred to females, but did not fertilize eggs [12,21]. It is still unclear if the elimination of sperm cells in *SD/SD⁺* males results from a failure in SNBP deposition in *SD⁺* spermatids, and more generally, what cytological events lead to *SD⁺* sperm elimination in the *SD* system.

Here we studied the histone-to-protamine transition in different strong *SD* genotypes with similar levels of segregation distortion. We show that the cytological phenotypes of *SD* males are variable, even though they share strong distortion strength, and this variability is largely associated with the number of *Rsp* repeats. Moreover, we show that the cytological phenotype can be profoundly modified by a genetic suppressor on the X chromosome.

## Results

### In *SD* males, nuclear defects are observed during the histone-to-protamine transition

To determine when *Rsp^s* gametes were eliminated in *SD/SD⁺* males, we chose to study spermatogenesis in two different genetic backgrounds that confer high distortion levels (see

Table 1 for *k* values, which are the ratio of flies carrying the *SD* chromosome over the total progeny of tested males). We used the $SD^+ cn^1 bw^1$ chromosome that we combined with the *SD-Mad* distorter chromosome, a strong distorter chromosome isolated from a wild population in Madison, WI [22]. We also used the $SD^+ Rsp^s$ dominantly marked balancer chromosome, *In(2LR)Gla* (hereafter *Gla*) [23], that we combined with the *SD5* distorter chromosome [22], another strong *SD* chromosome. To visualize the histone-to-protamine transition, we introduced a *protB-GFP* transgene—that expresses the *Drosophila* SNBP Protamine B (or Mst35Bb) fused to GFP—in the *SD* backgrounds. Both *Gla/SD5; protB-GFP* and $cn^1 bw^1/$*SD-Mad; protB-GFP* males (hereafter *Gla/SD5* and $cn^1 bw^1/SD$-*Mad*) harbored very strong distortion levels (*k* values 1 and 0.998, respectively, Table 1).

In *Drosophila*, spermatogenesis starts by the asymmetrical division of germ stem cells which give rise to spermatogonia. After 4 incomplete mitoses (without cytokinesis), each spermatogonium produces 16 interconnected spermatocytes that go through meiosis. Then each cyst of 64 spermatids differentiates into mature sperm cells in synchrony during spermiogenesis [24] (Fig 1B). We first observed the general organization of spermatogenesis in $SD/SD^+$ male testes. In both *SD* backgrounds, spermatogenesis appeared normal until the histone-to-protamine transition (Fig 1C), as previously reported [21]. During protamine incorporation, we observed many cysts containing both elongated spermatid nuclei and abnormally-shaped nuclei, often lagging behind. The abnormally-shaped nuclei also have loosely packed spermatid bundles indicating that they were eliminated before the end of spermiogenesis (Figs 1C and 2A). The abnormally shaped nuclei varied in appearance: some nuclei lost their elongated shape and were roundish, while some others were curled or crumpled. DNA-FISH analyses of *Gla/SD5* testes confirmed that the abnormally-shaped spermatid nuclei inherited the $SD^+$ chromosome with a large *Rsp* satDNA block and were eliminated (Fig 2B). However, we noticed that the phenotype of $cn^1 bw^1/SD$-*Mad* males was more variable. In fact, although the histone-to-protamine transition was perturbed, we observed the presence of some cysts with a less severe phenotype (Fig 2A). Moreover, the DNA-FISH *Rsp* probe stained both normally

**Table 1. Segregation distortion levels.** Unweighted means of *k* values ±standard deviation. n represents the number of flies in the offspring. For all male genotypes, *k* value is the ratio of offspring carrying the *SD* chromosome over the total number of flies except for $cn^1 bw^1/\underline{CyO}$*; protB-GFP* and *Gla/$\underline{CyO}$; protB-GFP* males. For these two genotypes, the *k* value represents the ratio of the offspring carrying the *CyO* chromosome over the total number of flies. The X chromosome carried by tested males is also indicated. The X chromosomes of the *RAL-380*, *RAL-313* and *RAL-309* strains carry a suppressor. A description of the crosses to obtain distorter and control genotypes are provided in S7 Fig.

| Genotype of fathers | *k* value ±SD | n | X chromosome |
|---|---|---|---|
| *SD-Mad/CyO; protB-GFP* | 0.443 ±0.074 | 1471 | $w^{1118}$ (from *Gla/CyO*) |
| *SD5/CyO; protB-GFP* | 0.338 ±0.076 | 1483 | $w^{1118}$ (from *Gla/CyO*) |
| $cn^1 bw^1/\underline{CyO}$*; protB-GFP* | 0.526 ±0.044 | 1403 | $y^1$ (from $cn^1 bw^1$) |
| *Gla/$\underline{CyO}$; protB-GFP* | 0.505 ±0.051 | 2530 | $w^{1118}$ (from *Gla/CyO*) |
| $cn^1 bw^1/SD$-*Mad; protB-GFP* | 0.998 ±0.005 | 705 | $y^1$ (from $cn^1 bw^1$) |
| *Gla/SD5; protB-GFP* | 1 ±0.00 | 938 | $w^{1118}$ (from *Gla/CyO*) |
| *Gla/SD-Mad; protB-GFP* | 0.999 ±0.004 | 1214 | $w^{1118}$ (from *Gla/CyO*) |
| $cn^1 bw^1/SD5$*; protB-GFP* | 0.999 ±0.003 | 1012 | $y^1$ (from $cn^1 bw^1$) |
| *RAL-313/Cy SD-Mad* | 0.975 ±0.013 | 2579 | X from *Cy SD-Mad* |
| *RAL-313/Cy SD-Mad* | 0.757 ±0.071 | 1622 | X from *RAL-313* |
| *RAL-309/Cy SD-Mad* | 0.956 ±0.058 | 2270 | X from *Cy SD-Mad* |
| *RAL-309/Cy SD-Mad* | 0.927 ±0.067 | 2228 | X from *RAL-309* |
| *RAL-380/Cy SD-Mad* | 0.997 ±0.004 | 2598 | X from *Cy SD-Mad* |
| *RAL-380/Cy SD-Mad* | 0.941 ±0.031 | 2185 | X from *RAL-380* |

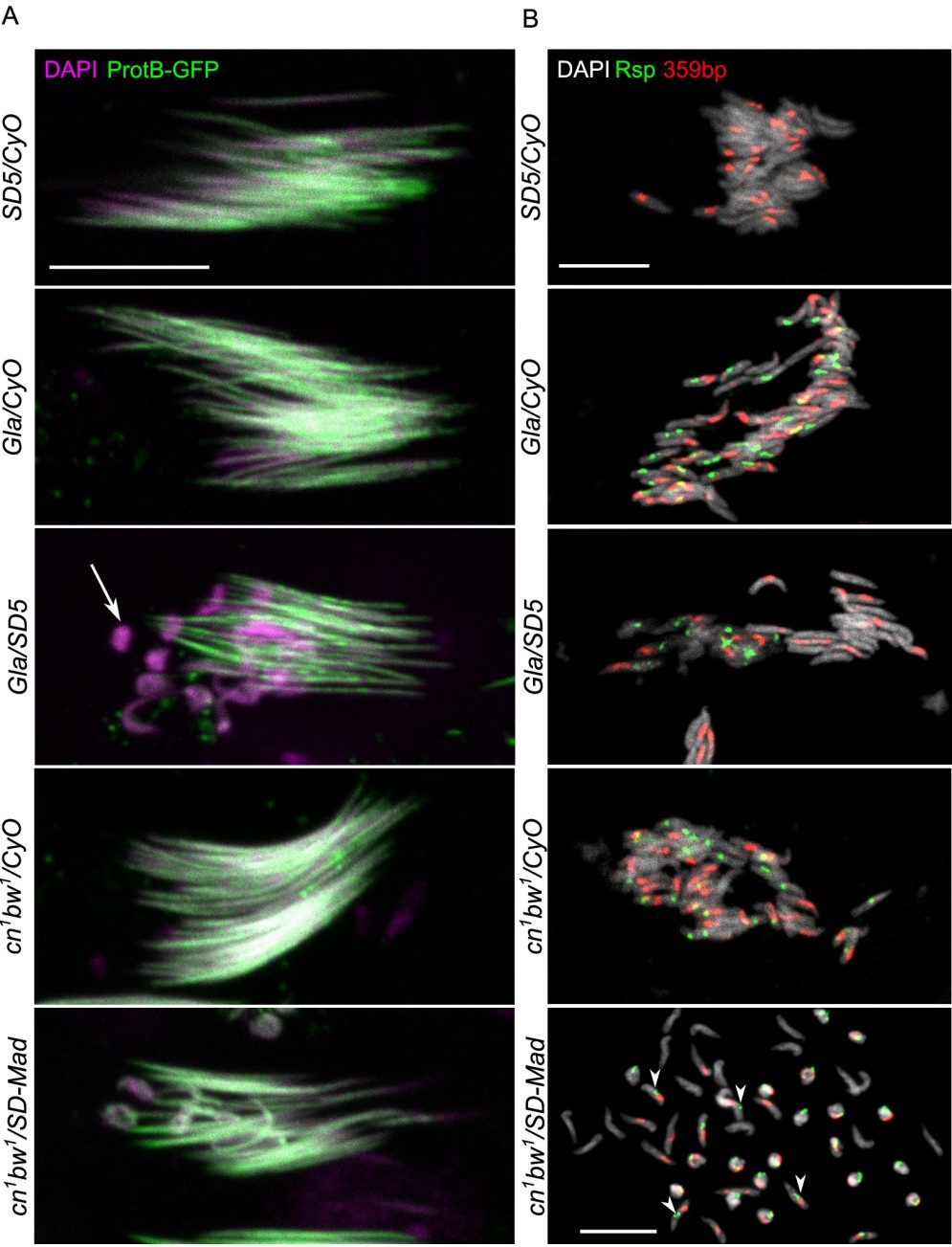

**Fig 2. *Rsp^s* nuclei are eliminated during the histone-to-protamine transition in *SD* males.** (**A**) Confocal images of individual cysts of 64 spermatids in males carrying the *protB-GFP* transgene and stained with DAPI (magenta). In *Gla/SD5; protB-GFP* testes, about half spermatid nuclei appear abnormally shaped and are eliminated (arrow). In *cn¹ bw¹/SD-Mad*, the number of abnormally-shaped nuclei is often lower than in *Gla/SD5*. Scale bar: 10μm. (**B**) DNA-FISH staining of squashed testes with specific probes for *Rsp* (green) and *359 bp* satDNA on the X chromosome (red). In *Gla/SD5*, the abnormally-shaped spermatid nuclei that are eliminated carry the second chromosome with the large *Rsp* satDNA block. In *cn¹ bw¹/SD-Mad*, both abnormally-shaped and normally-shaped (arrowheads) spermatid nuclei carry a second chromosome with a large *Rsp* satDNA block. Scale bar: 10μm. Note that nuclear length and width in FISH experiments can be slightly altered by the experimental procedure (squash, treatments to remove proteins and denature DNA).

and abnormally-shaped spermatid nuclei (Fig 2B). These observations suggested that the mode of spermatid elimination may differ between *cn¹ bw¹/SD-Mad* and *Gla/SD5* genotypes.

## Histone elimination and Tpl94D transient incorporation are slightly delayed in *Gla/SD5* males

To determine the precise stage of *SD⁺* spermatid nuclei elimination and the defects that these nuclei may have, we analyzed the histone-to-protamine transition in *SD* males in detail. We first stained testes for histones and the transition protein Tpl94D. In *cn¹ bw¹/SD-Mad* testes, the dynamics of histone elimination and transient Tpl94D expression were comparable to *Gla/CyO* (Fig 3) and *SD5/CyO* (S1A Fig) control males. Histone signals progressively decreased as Tpl94D signal increased in elongating spermatids. Then, the Tpl94D signal also vanished and at the end of spermiogenesis, histones and Tpl94D were undetectable in all nuclei. In *Gla/SD5* males, we also observed progressive histone elimination, transient expression of Tpl94D, and the eliminated spermatid nuclei were all negative for histones and Tpl94D. However, we repeatedly observed some spermatid nuclei (ca. 4–8 per cyst) with a faint histone signal in cysts that were also positive for Tpl94D (Fig 3, white arrows and see oversaturated image). At this stage, histones were barely detected in control and *cn¹ bw¹/SD-Mad* flies. We also observed that about half of the nuclei showed a weaker Tpl94D signal than the other half (Fig 3). These

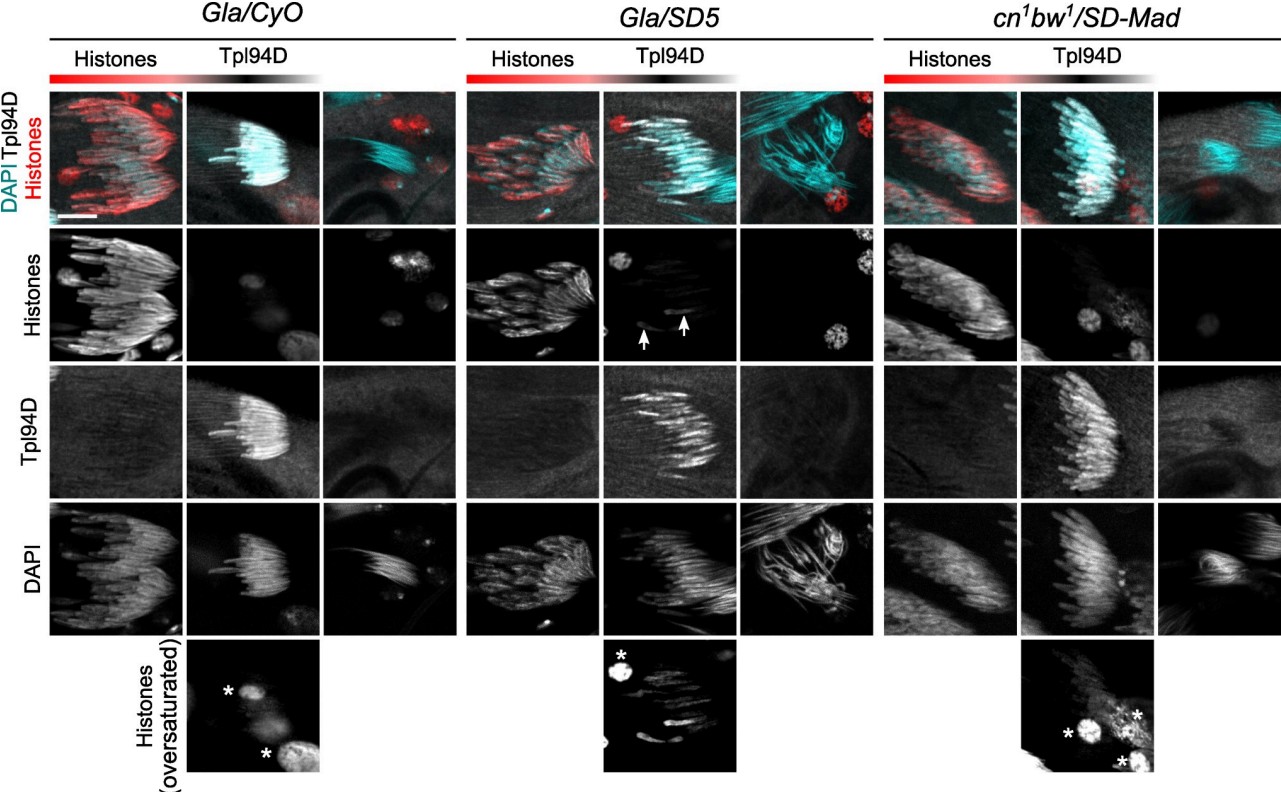

**Fig 3. Histone elimination and Tpl94D transient expression are slightly delayed in *Gla/SD5* males.** Confocal images of whole-mount testes from males stained with a pan-histone antibody (red), an antibody against the Tpl94D transition protein (white) and DAPI (cyan). Each image shows one or two cysts of 64 spermatid nuclei at the indicated stage of the histone-to-protamine transition that was estimated with nuclear shape and staining intensity for each signal. In control *Gla/CyO* males, spermatid nuclei which are positive for Tpl94D have almost lost all their histones (see oversaturated image on bottom panels, asterisks indicate somatic nuclei). At the end of spermiogenesis, all spermatid nuclei are negative for Tpl94D and histones. In *Gla/SD5* spermatids, traces of histones are detected in about half the nuclei in cysts that have incorporated Tpl94D (arrows, and oversaturated image below). Tpl94D staining is weaker in about half of the nuclei, compared to sister nuclei. However, at the end of spermiogenesis, all spermatid nuclei are negative for both histones and Tpl94D. In *cn¹ bw¹/SD-Mad*, histone elimination and transient Tpl94D expression appear normal. Scale bar: 10μm.

observations suggested that histone elimination and transient Tpl94D expression were delayed in $SD^+$ spermatid nuclei in *Gla/SD5* males but not in *cn$^1$ bw$^1$/SD-Mad* males.

## Protamine incorporation is incomplete in $SD^+$ nuclei of *Gla/SD5* males and stops prematurely

We then studied progression of protamine incorporation in *Gla/SD5* and *cn$^1$ bw$^1$/SD-Mad* testes using the *protB-GFP* transgene. To determine more precisely the stage of spermatid differentiation, we stained testes with phalloidin to reveal individualization complexes (IC). IC are actin cones that form around spermatid nuclei after the histone-to-protamine transition and mark the onset of individualization. During this process, IC move along spermatid axes to remove excess materials and cytoplasm and invest each of the 64 interconnected spermatids with their own cell membrane (Fig 4A) [13,25].

In *Gla/SD5* testes, all nuclei in cysts of early elongating spermatids were normally shaped and had started to incorporate protamines. However, about half of the nuclei showed weaker ProtB-GFP signals compared to the other half (pre-IC in Fig 4B, Fig 4C and see *SD5/CyO* control in S1B Fig). Then, when individualization started, the weakly stained nuclei in *Gla/SD5* appeared larger than their sister nuclei (IC, Fig 4B). At the end of spermiogenesis, the $SD^+$ nuclei were abnormally shaped, weakly stained with ProtB-GFP and lagged behind the rest of the bright ProtB-GFP positive, needle-shaped nuclei (post-IC, Fig 4B). This suggested that in *Gla/SD5* males, $SD^+$ spermatid nuclei incorporated fewer SNBPs and were eliminated during individualization. This phenotype was comparatively weaker and more variable in *cn$^1$ bw$^1$/SD-Mad* testes. In each cyst, we observed weakly stained ProtB-GFP (pre-IC) and abnormally shaped spermatid nuclei (IC and post-IC) but they were infrequent and varied in number (Fig 4B). These observations suggested that in *cn$^1$ bw$^1$/SD-Mad*, the histone-to-protamine transition was generally less disturbed in $SD^+$ nuclei and a significant fraction of these nuclei progressed normally through this important chromatin transition.

## Chromatin compaction is abnormal in $SD^+$ spermatids in $SD/SD^+$ males

Previous electron microscopy studies reported that half of nuclei presented chromatin condensation defects within the same cyst of elongating spermatids from $SD/SD^+$ males [12]. We thus examined chromatin compaction using an antibody that recognizes double-stranded DNA (anti-dsDNA) to take advantage of the reduced accessibility to spermatid DNA after chromatin compaction. Indeed, in sperm heads, chromatin is so tightly compacted, that nuclei are refractory to antibody staining [26]. As expected, in wildtype testes, the anti-dsDNA antibody stained somatic and germinal nuclei except IC and post-IC stage spermatids and sperm nuclei (S2 Fig). Moreover, the staining strength was directly and negatively linked to the level of chromatin compaction. We then stained control and $SD/SD^+$ testes with the anti-dsDNA antibody. In control *Gla/CyO* and *SD5/CyO* males, all spermatid nuclei in pre-IC cysts were evenly stained with the anti-dsDNA antibody. As protamine incorporation and chromatin compaction progressed, this staining vanished (Figs 5 and S1C). In contrast, in *Gla/SD5* testes, the anti-dsDNA staining appeared stronger in about half of the spermatid nuclei in pre-IC cysts (Fig 5). These nuclei corresponded to the ones that were weakly stained with ProtB-GFP, supporting the hypothesis that $SD^+$ nuclei failed to fully incorporate SNBPs and compact their chromatin properly. Moreover, at later stages (IC and post-IC, Fig 5), the eliminated nuclei were brightly stained with the anti-dsDNA antibody, whereas the nuclei that differentiated normally were all negative for this marker. In *cn$^1$ bw$^1$/SD-Mad* testes, this phenotype was again weaker and highly variable. Notably, in IC and post-IC cysts, abnormally-shaped spermatid nuclei brightly stained with the anti-dsDNA antibody were less frequent. However, we

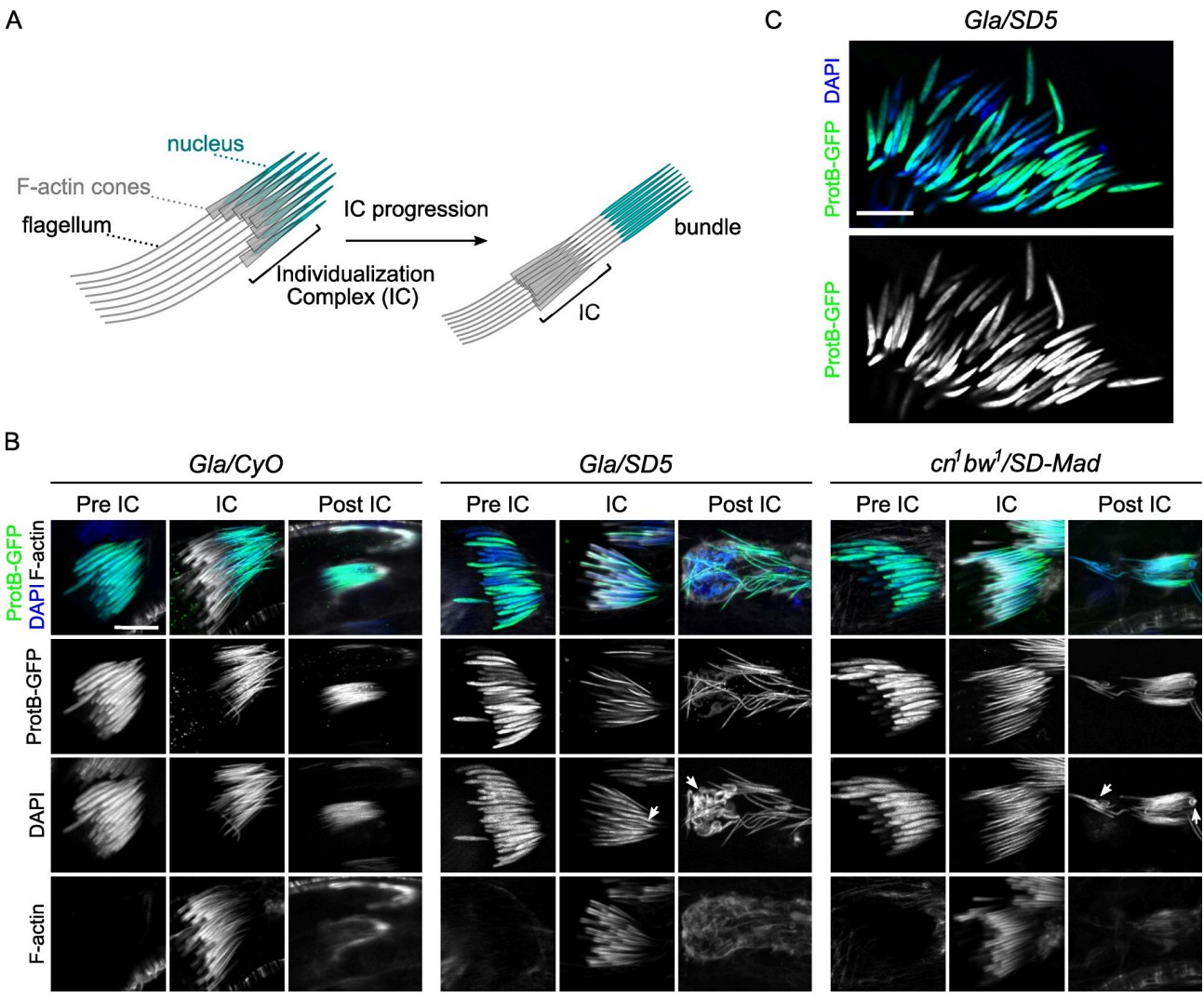

**Fig 4. Protamine incorporation is delayed in *Gla/SD5* and slightly disturbed in *cn¹ bw¹/SD-Mad* testes.** (**A**) A scheme of spermatid individualization. At the end of histone-to-protamine transition, actin cones form around each spermatid nuclei (individualization complex, IC) and progress along the flagellum to remove cytoplasmic excess. (**B**) Confocal images of whole-mount testes from males carrying a *protB-GFP* transgene (green) and stained with DAPI (blue) and phalloidin (F-actin, white), that reveals individualization complex (IC). Each square shows a cyst of 64 spermatid nuclei at the indicated stage [before individualization (pre-IC), at the onset of individualization (IC) and after individualization (post-IC)]. In control *Gla/CyO* males, ProtB-GFP fluorescence and nuclear shape appear homogeneous for the 64 spermatids at all stages. In *Gla/SD5* testes, about half of the nuclei show weaker ProtB-GFP signals in pre-IC cysts although nuclear shapes appear similar with DAPI staining. During individualization, the nuclei that present weaker GFP signals also appear larger (arrow). In post-IC stages, these nuclei are abnormally shaped (arrow) and eliminated in the waste bag. In *cn¹ bw¹/SD-Mad* testes, this phenotype is weaker. In pre-IC and IC spermatids, ProtB-GFP signals are more homogeneous. In post-IC stages, only a few nuclei seem to be abnormally shaped (arrows). This phenotype is variable from one cyst to another and a *Gla/SD5*-like phenotype can be occasionally observed. Scale bar: 10μm. (**C**) A magnified view of a pre-IC cyst in a *Gla/SD5* testis stained with DAPI showing a weaker ProtB-GFP signal in about half of spermatids. Scale bar: 10μm.

also observed needle-shaped nuclei weakly stained with the anti-dsDNA antibody that were included in the bundles of individualized spermatids (Fig 5).

## Seminal vesicles of *cn¹ bw¹/SD-Mad* males contain abnormally condensed sperm nuclei

Our previous observations of *cn¹ bw¹/SD-Mad* testes suggested that abnormally compacted *SD⁺* nuclei escaped elimination during individualization and differentiated into mature

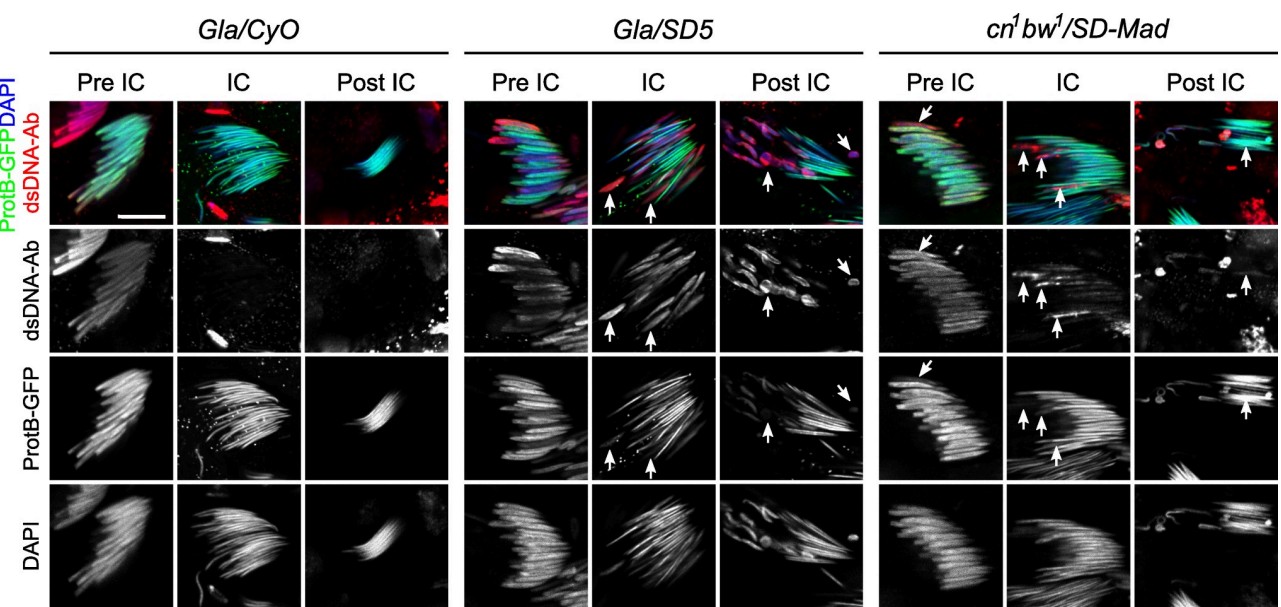

**Fig 5. *Rsp^s* spermatid nuclei are abnormally compacted in both *SD* genotypes.** Confocal images of whole-mount testes from males carrying the *protB-GFP* transgene stained with an anti-dsDNA antibody (dsDNA-Ab; red) to probe for chromatin compaction, DAPI (blue) and phalloidin (not shown for clarity). In *Gla/CyO* control males, all nuclei in pre-IC cysts are homogenously and weakly stained with the anti-dsDNA antibody. When individualization starts (IC) and after (post-IC), spermatid nuclei are not stained with the anti-dsDNA antibody because the highly compacted chromatin becomes inaccessible to antibodies. In *Gla/SD5* pre-IC cysts, about half of the nuclei are more brightly stained with the anti-dsDNA antibody compared to their sister nuclei. This strong anti-dsDNA staining is negatively correlated to ProtB-GFP intensity signals. During individualization (IC), the abnormally shaped nuclei that are weakly stained with ProtB-GFP are also positively and brightly stained with the anti-dsDNA antibody (arrows). After individualization, the eliminated nuclei are also brightly stained with the anti-dsDNA antibody (arrows) whereas needle-shaped nuclei remain negative for this staining. In *cn^1 bw^1/SD-Mad* testes, spermatid nuclei in pre-IC cysts are homogeneously stained with the anti-dsDNA antibody except for a few nuclei which are more brightly stained (arrow). However, during individualization, some spermatid nuclei are brightly stained with the anti-dsDNA antibody (arrows, IC) and about half of the nuclei show a faint anti-dsDNA signal suggesting that most *SD^+* nuclei are not normally compacted. Cysts of post-IC spermatids contain both abnormally-shaped and needle-shaped anti-dsDNA positive nuclei (arrow, post-IC). These latter nuclei are included in the bundle of *SD* spermatids, suggesting that they are not eliminated. Scale bar: 10μm.

spermatozoa. To test this hypothesis, we examined seminal vesicle contents stained with the anti-dsDNA antibody. As expected, in *cn^1 bw^1/CyO* and *Gla/CyO* control males, seminal vesicles were filled with needle-shaped sperm nuclei brightly stained with ProtB-GFP that were almost all negative for the anti-dsDNA antibody (Figs 6A and S3A). In *Gla/SD5* males, seminal vesicles were smaller and contained fewer nuclei than control seminal vesicles. In these seminal vesicles, nearly all sperm nuclei were needle-shaped, brightly stained with ProtB-GFP and negative for the anti-dsDNA antibody. In striking contrast, *cn^1 bw^1/SD-Mad* seminal vesicles contained many anti-dsDNA positive sperm nuclei (Fig 6A). Some of these nuclei were abnormally shaped but most of them were needle shaped, suggesting that although the *SD^+* nuclei acquired an elongated shape, their chromatin was not properly compacted. To verify that these nuclei corresponded to *SD^+ Rsp^s* nuclei, we performed DNA-FISH on squashed seminal vesicles with a *Rsp* probe and a control probe specific to the *359 bp* satDNA, a large X-linked satDNA block (Figs 6B and S3B). In *Gla/SD5* male vesicles, about half of nuclei were stained with the *359 bp* probe as expected, while *Rsp* positive nuclei were very rare or absent, confirming that nearly all *SD^+ Rsp^s* spermatids were eliminated during spermiogenesis. On the contrary, in *cn^1 bw^1/SD-Mad* males, many sperm nuclei were stained with the *Rsp* probe, confirming that *SD^+ Rsp^s* nuclei were released in the seminal vesicle.

Image analysis and quantification revealed that seminal vesicles contained about 39% and 2% of anti-dsDNA positive sperm nuclei in *cn^1 bw^1/SD-Mad* and *Gla/SD5* males, respectively

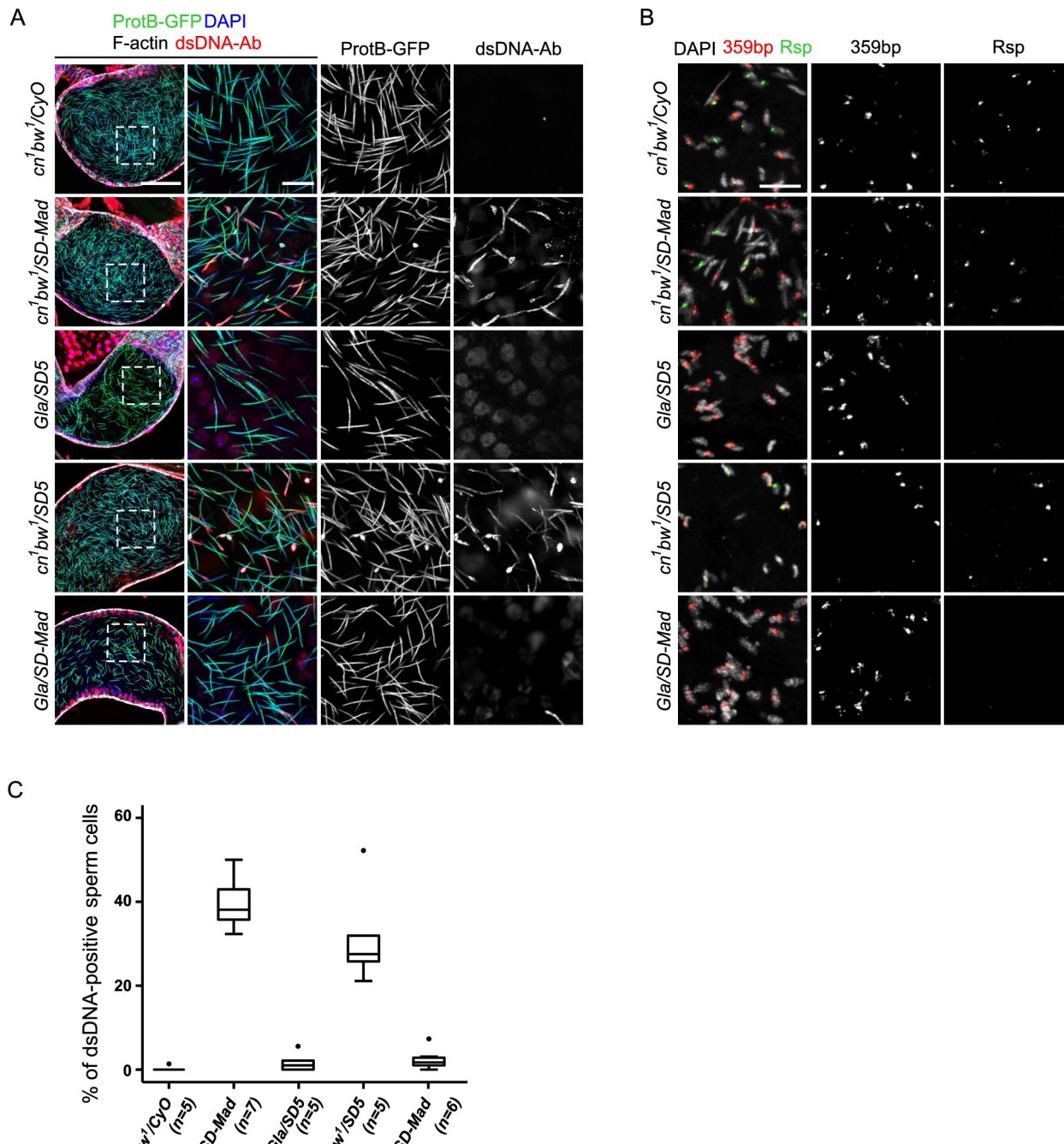

**Fig 6. Seminal vesicles of *cn¹ bw¹ SD* males contain many abnormally compacted *Rspˢ* nuclei.** (**A**) Confocal images of whole-mount seminal vesicles from males of the indicated genotype, carrying the *protB-GFP* transgene (green) and stained with the anti-dsDNA antibody (dsDNA-Ab; red), DAPI (blue) and phalloidin (white). A wide view of the seminal vesicle is shown on left panels (scale bar: 50μm). Dashed white squares correspond to magnified regions shown on right panels (scale bar: 10μm). In *cn¹ bw¹/CyO* seminal vesicles, nearly all sperm nuclei are negative for the anti-dsDNA antibody. Seminal vesicles from *cn¹ bw¹/SD-Mad* and *cn¹ bw¹/SD5* males contain many anti-dsDNA positive sperm nuclei, suggesting that they are abnormally condensed although most of them are needle-shaped and contain protamines. In *Gla/SD5* and *Gla/SD-Mad* seminal vesicles, such nuclei are rarely observed. Note that the seminal vesicles of *SD* males are smaller in general but *Gla/SD5* and *Gla/SD-Mad* vesicles are even smaller than *cn¹ bw¹/SD5 and cn¹ bw¹/SD-Mad* ones, indicating that they contain less sperm. (**B**) DNA-FISH performed on seminal vesicle contents of the indicated genotype with a *Rsp* probe (green) and a control probe for the *359 bp* satDNA (red) on the X chromosome as control (scale bar: 10μm). In all panels, sperm nuclei have been partially decondensed with DTT to facilitate probe penetration. In control *cn¹ bw¹/CyO* seminal vesicles, about half of the nuclei are positive for the *Rsp* probe demonstrating that *SD⁺ Rspˢ* spermatozoa are normally produced.

In $cn^1$ $bw^1$/SD-Mad and $cn^1$ $bw^1$/SD5 seminal vesicles, many nuclei are positive for the *Rsp* probe, thus confirming that the anti-dsDNA positive sperm nuclei detected in seminal vesicles of these genotypes correspond to abnormal spermatozoa that inherited the $SD^+$ $Rsp^s$ chromosome. (**C**) Box plot of the percentage of anti-dsDNA positive sperm nuclei in a Z-stack square as shown in middle panels in A. Whiskers show minimum and maximum values, boxes show the middle 50% of the values and horizontal lines represent medians. Black dots indicate outliers. For each genotype, n indicates the number of males, one seminal vesicle was analyzed per male.

(n = 7 and n = 5, respectively, n is the number of males (one seminal vesicle analyzed per male); Fig 6C). This quantification showed that in *Gla/SD5* males, less than one $SD^+$ nucleus of 32 $SD^+$ expected nuclei per cyst on average was released in the seminal vesicle, whereas, in $cn^1$ $bw^1$/SD-Mad males, 20.5 of 32 $SD^+$ nuclei per cyst on average were released. From our previous observation, we suspected that the number of eliminated nuclei was variable from one cyst to another in $cn^1$ $bw^1$/SD-Mad males. In support of this observation, Tokuyasu and colleagues reported that in $cn^1$ $bw^1$/SD72 males, the number of abnormal spermatids that failed to individualize in a cyst varied from 0 to 32 but that the number of spermatids with incompletely condensed chromatin was nearly 32 [27]. Taken together, our results showed that the cytological phenotypes of two $SD/SD^+$ male genotypes with very high and comparable distortion levels were largely different. In *Gla/SD5* males, $SD^+$ spermatids are systematically eliminated during individualization whereas, in $cn^1$ $bw^1$/SD-Mad males, $SD^+$ abnormal sperm cells are released in the seminal vesicle. Importantly, since the $cn^1$ $bw^1$/SD-Mad males exhibited very high levels of segregation distortion (*k* value 0.998; Table 1), we inferred that these abnormally condensed sperm cells were unable to fertilize eggs. In fact, anti-dsDNA positive sperm nuclei were only very rarely observed in the sperm storage organs of females mated to $cn^1$ $bw^1$/SD-Mad males (S4 Fig).

We then wondered whether the difference between *SD* male cytological phenotypes was linked to the $SD^+$ $Rsp^s$ or the *SD* chromosome. We thus set up reciprocal crosses to generate $cn^1$ $bw^1$/SD5 and *Gla/SD-Mad* males and used DNA-FISH and the anti-dsDNA antibody to detect $SD^+$ $Rsp^s$ sperm nuclei in seminal vesicles. While we observed many $SD^+$ $Rsp^s$ escaper nuclei in the seminal vesicles of $cn^1$ $bw^1$/SD5 males, we seldom detected $SD^+$ $Rsp^s$ nuclei in *Gla/SD-Mad* seminal vesicles (Fig 6). Moreover, during the histone-to-protamine transition, half of spermatid nuclei in *Gla/SD-Mad* incorporated fewer protamines, were abnormally compacted and eliminated during individualization, similar to what we observed in *Gla/SD5* males (S5 Fig). On the contrary, in $cn^1$ $bw^1$/SD5 testes, this phenotype was weaker and variable, as we previously observed in $cn^1$ $bw^1$/SD-Mad (S5 Fig). These results thus support the hypothesis that the difference between the two *SD* cytological phenotypes depends on the $SD^+$ $Rsp^s$ chromosome.

## Systematic $SD^+$ spermatid elimination during individualization occurs when *Rsp* carries 2000 copies or more

Since the strength of *SD* male segregation distortion positively correlates with *Rsp* copy number [9,10], we wondered whether this factor could account for the observed difference between cytological phenotypes. The structure of the *Rsp* satDNA locus on the $cn^1$ $bw^1$ chromosome carried by the *iso-1* strain has been characterized in detail with single-molecule long-read sequencing and validated with molecular and computational approaches [28]. The major *Rsp* locus of *iso-1* flies contained ≈1050 *Rsp* repeats spread across ≈170 kb. However, the copy number and the organization of the *Rsp* satDNA of the *Gla* chromosome, is unknown. We thus estimated the number of *Rsp* satDNA copies on the *Gla* chromosome relative to $cn^1$ $bw^1$ chromosome by quantitative PCR on genomic DNA using two sets of primers that we designed using the published canonical *Rsp* left and right sequences [28]. We used $cn^1$ $bw^1$/

*SD-Mad* flies to normalize the quantification and validated by quantifying copy number in *SD-Mad* and $cn^1$ $bw^1$ homozygous flies. Our results showed that *SD-Mad/SD-Mad* carried about 20 copies consistent with previous studies [9] and $cn^1$ $bw^1$ homozygous flies carried ≈2000 copies, as expected. Interestingly, the quantification of *Gla/SD-Mad* flies revealed that the *Gla* chromosome carried 2800–3800 copies, depending on the primer set (Fig 7A). This variation may be due to the organization of the *Rsp* satDNA on the *Gla* chromosome. Indeed, although the canonical *Rsp* repeat is a dimer of related left and right *Rsp* sequences, tandem monomeric repeats (e.g. multiple right *Rsp*) are occasionally interspersed with the dimers. Therefore, the number of left and right *Rsp* in a locus can vary. Our quantification nevertheless suggested that the *Gla* chromosome contained at least twice as many repeats as the $cn^1$ $bw^1$ chromosome. To verify that the *Gla* chromosome carried a larger *Rsp* satDNA block, we also performed DNA-FISH on squashed spermatid nuclei (S6A Fig). Consistent with our expectations, *Rsp* fluorescent signals were larger in *Gla* spermatid nuclei compared to $cn^1$ $bw^1$ (S6B Fig). These results thus support our hypothesis that the cytological phenotype is linked to *Rsp* copy number.

To further challenge our hypothesis, we studied the phenotype of other *SD/SD+* males bearing the *SD-Mad* chromosome combined with *SD+* chromosomes carrying different *Rsp* copy numbers. We selected three strains from the Drosophila Genetics Reference Panel (DGRP)—a collection of sequenced inbred *D. melanogaster* lines derived from wild-caught flies [29,30]—that carried different *Rsp* copy numbers that we estimated by qPCR: *RAL-313* (≈1000 copies), *RAL-309* (≈1300 copies) and *RAL-380* (≈2500–2700 copies) (Fig 7A). We also performed DNA-FISH on spermatid nuclei and quantified the size of *Rsp* signals relative to nuclear size. The relative size of *Rsp* signals was also higher in *RAL-380* and smaller in *RAL-313* thus corroborating the qPCR results (S6B Fig). We crossed these lines to *SD-Mad; protB-GFP* flies to examine testes and seminal vesicles of the male progeny. Interestingly, the cytological phenotype of *RAL-380/SD-Mad* males was similar to *Gla/SD5*: in pre-IC cysts, about half of spermatid nuclei showed a weaker ProtB-GFP signal and stronger anti-dsDNA staining compared to the other half (Fig 7B). In IC and post-IC cysts, these nuclei were abnormally shaped and compacted. Finally, seminal vesicles contained almost no anti-dsDNA positive sperm nuclei (Fig 7B). In contrast, the distorter males that carried *SD+* chromosomes from the *RAL-309* and *RAL-313* strains, that contained fewer *Rsp* copies than *RAL-380*, showed a phenotype similar to $cn^1$ $bw^1$/*SD-Mad* and $cn^1$ $bw^1$/*SD5* males (Fig 7B). These results confirmed that the phenotype of systematic spermatid elimination before the release in seminal vesicles is associated with a very high *Rsp* copy number. They also suggested a threshold for *Rsp* copy number (>2000) above which spermatid elimination is systematic during individualization.

## A suppressor on the X modifies the phenotype of spermatid elimination

In the course of our experiments, we noticed that distortion levels were lower when we used females, instead of males, from the *RAL-313*, *RAL-309* and *RAL-380* strains in our crosses to produce *SD/SD+* males (see *k* values in Table 1 and S7C and S7D Fig for a description of the crosses). This observation suggested that the X chromosomes of these strains carried one or more genetic elements that act as a weak suppressor of *SD* (hereafter *Su(SD)X-380*, *Su(SD)X-309* and *Su(SD)X-313*). To study the impact of these suppressors on spermiogenesis progression, we generated *SD/SD+* males bearing *Su(SD)X* and the *protB-GFP* transgene and stained their testes with DAPI and phalloidin. Strikingly, in males that carried an *SD+* chromosome with >2000 copies (*RAL-380*), the cytological phenotype was profoundly modified by the presence of *Su(SD)X-380*. Indeed, whereas seminal vesicles of $w^{1118}$; *RAL-380/SD-Mad* males contained <2% of anti-dsDNA positive nuclei, we counted up to ≈30% of such nuclei in *Su(SD)*

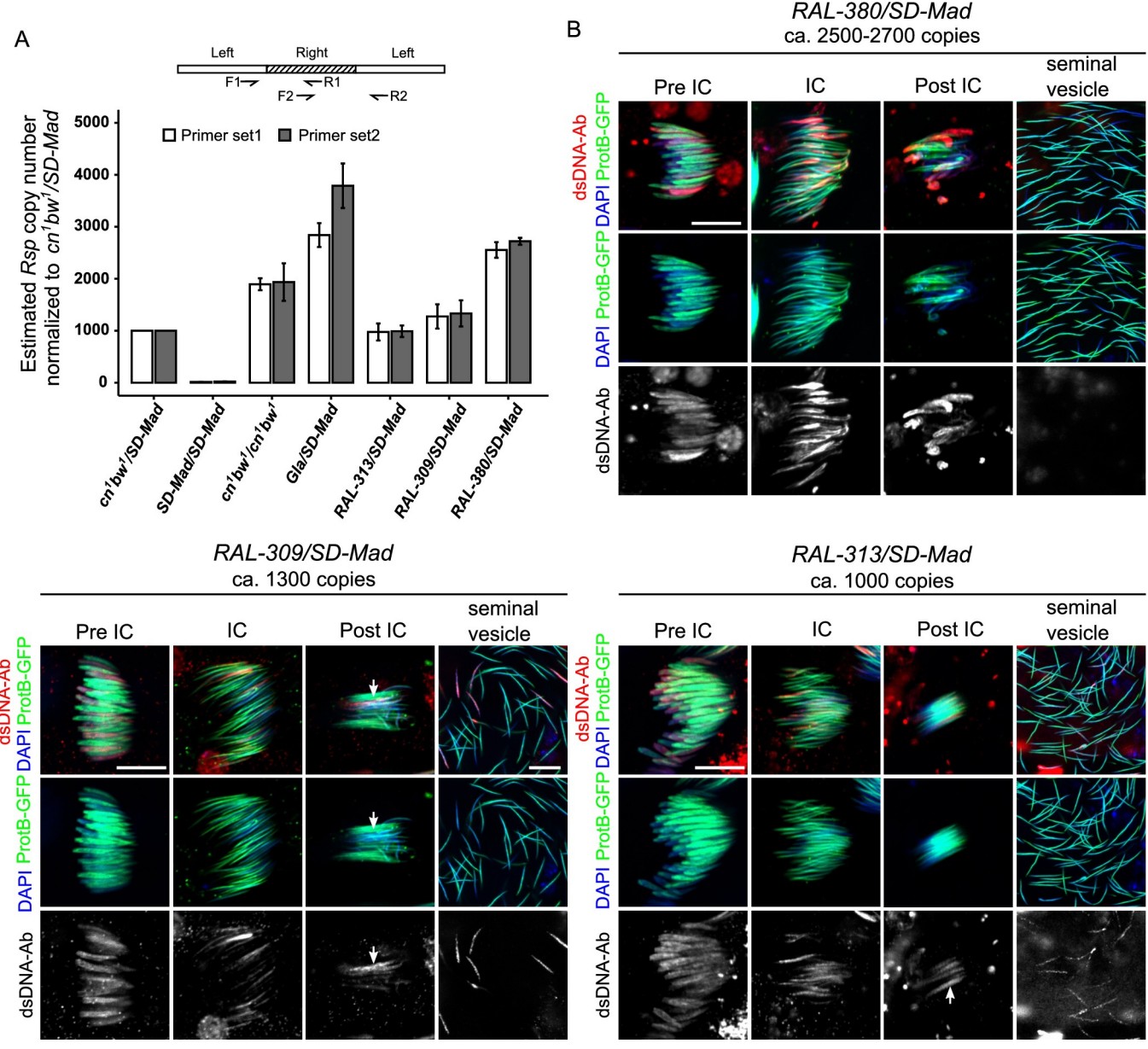

**Fig 7. The cytological phenotype of *SD* males is linked to *Rsp* copy numbers.** (**A**) *Rsp* copy numbers on the *Gla* chromosome and the second chromosome of *RAL* strains were estimated by qPCR on genomic DNA. For the quantification, two sets of *Rsp* primers were designed using the canonical *Rsp* left and right sequences published in [28] (see primer positions on the scheme above histograms). The copy number in *cn¹ bw¹/SD-Mad* was set to 1000 based on [28]. (**B**) The cytological phenotype of *RAL-380/SD-Mad* (top right panels), *RAL-309/SD-Mad* (bottom left panels), and *RAL-313/SD-Mad* (bottom right panels) testes carrying the *protB-GFP* transgene and stained with an anti-dsDNA antibody (dsDNA-Ab; red), phalloidin (not shown for clarity) and DAPI (blue). *RAL-380/SD-Mad* which carry ca. 2500–2700 *Rsp* copies present a phenotype similar to *Gla/SD5* males. About half of the nuclei incorporate less protamines and are abnormally compacted as revealed by the bright anti-dsDNA staining (pre-IC and IC cysts). These nuclei are eliminated during individualization (IC and post-IC) and seminal vesicles contain almost no abnormal anti-dsDNA positive nuclei. In *RAL-309/SD-Mad* (ca. 1300 *Rsp* copies), although protamine incorporation is also delayed in half of nuclei (pre-IC), many anti-dsDNA positive nuclei are detected in the bundle of spermatid after individualization (arrow). These abnormally condensed nuclei are detected in seminal vesicles. In *RAL-313/SD-Mad* males (ca. 1000 *Rsp* copies), protamine incorporation appears less disturbed but many anti-dsDNA positive nuclei are detected during individualization, in the bundle of spermatid nuclei after individualization (arrow) and in seminal vesicles. Scale bar: 10μm.

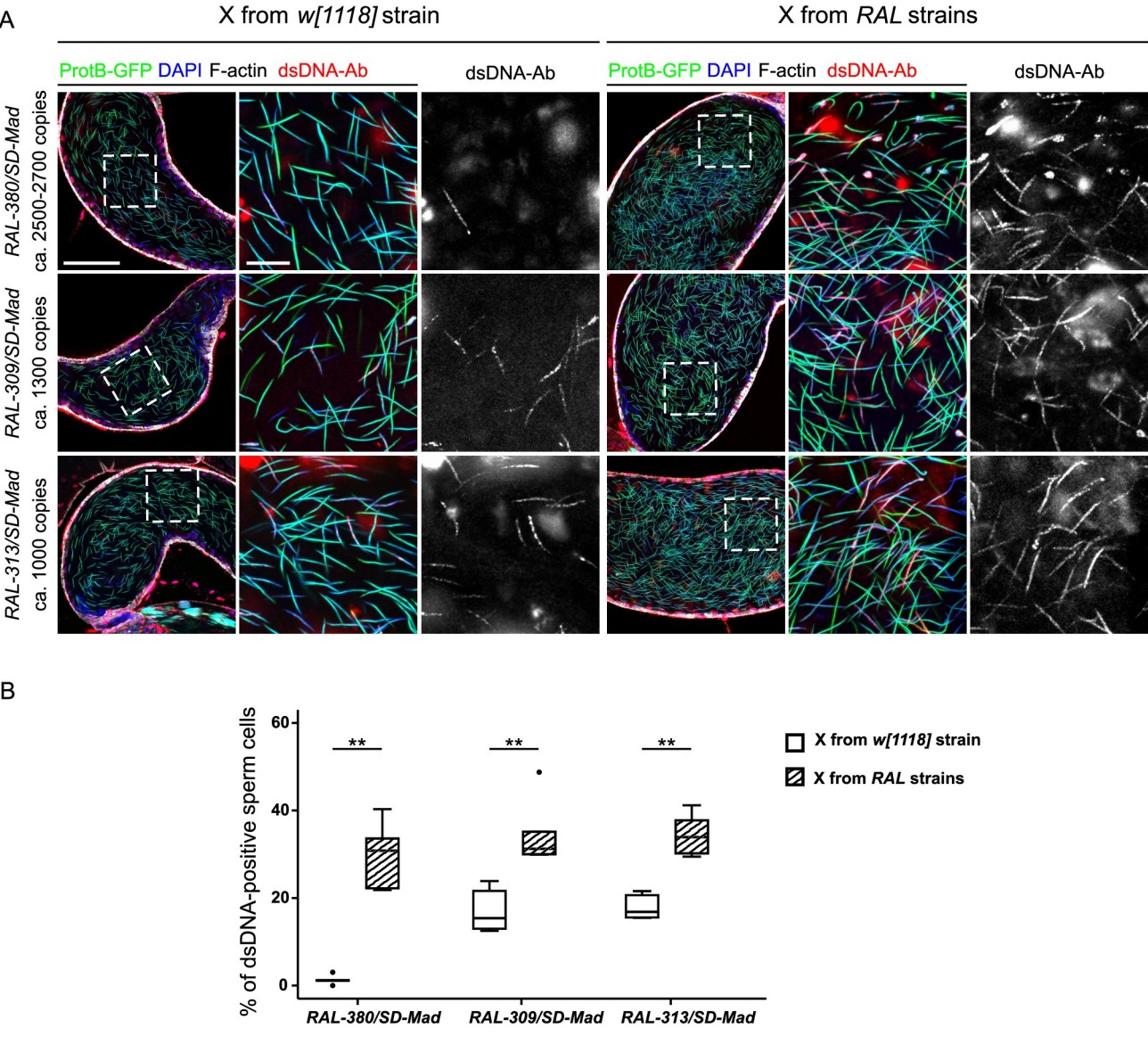

**Fig 8. *SD* male cytological phenotypes are modified by a suppressor on the X chromosome. (A)** Confocal images of seminal vesicles stained with an anti-dsDNA antibody (dsDNA-Ab; red), phalloidin (white) and DAPI (blue) from *RAL-380/SD-Mad*, *RAL-309/SD-Mad*, and *RAL-313/SD-Mad* males carrying the *protB-GFP* transgene with the X chromosome from the *w1118; SD-Mad/CyO; protB-GFP* strain, which does not carry a suppressor (left panels) or from the *RAL* strains which carries *Su(SD)X* (right panels). Scale bars: 50μm for large views of the seminal vesicles and 10μm for zoom-ins. **(B)** A box plot showing a quantification of the percentage of anti-dsDNA positive sperm nuclei in seminal vesicles of the genotypes shown in (A). Whiskers show minimum and maximum values, boxes show the middle 50% of the values and horizontal lines represent medians. For each genotype, five seminal vesicles from five different males were analyzed. Wilcoxon test, ** p-value<0.01.

*X-380*; *RAL-380/SD-Mad* (Fig 8). Moreover, the histone-to-protamine transition in *Su(SD)X-380*; *RAL-380/SD-Mad* males appeared less disturbed (S8 Fig). It thus appeared that the presence of *Su(SD)X-380* might allow for the release of many *SD+* nuclei in the seminal vesicles. However, the strong segregation distortion (*k* value 0.941 with *Su(SD)X-380* versus 0.997 without; Table 1) suggested that most of these nuclei were not functional. In the *RAL-313* and *RAL-309 SD/SD+* males, *Su(SD)X* had no significant effect on spermiogenesis progression. However, the proportion of anti-dsDNA positive nuclei in the seminal vesicles of *Su(SD)X-313*; *RAL-313/SD-Mad* and *Su(SD)X-309*; *RAL-309/SD-Mad* males increased by ≈15 and

≈18%, respectively (Fig 8). These results indicated that some X-linked suppressors modified spermatid elimination efficiency at individualization. Thus, a modest effect on segregation distortion levels could underly substantial modifications of the cytological phenotype.

## Discussion

Our study shows that the cytological phenotypes of strong distorter males (*k* value> 0.95) can be classified into at least two categories. In the first category, *SD*⁺ spermatid nuclei show a delay in the initial steps of the histone-to-protamine transition, followed by a premature arrest of protamine incorporation, abnormal nuclear condensation and systematic elimination during individualization. In the second category, *SD*⁺ spermatid nuclei display fewer disturbances of the histone-to-protamine transition: while protamine incorporation is perturbed in some nuclei, many *SD*⁺ spermatids progress through spermiogenesis and are eventually released in the seminal vesicles. Our study reveals for the first time that these mature *SD*⁺ spermatozoa have improperly compacted nuclei despite their apparent normal shape.

A remarkable and well-established feature of *SD* is the positive correlation between *Rsp* copy number and sensitivity to distortion [9,10]. Our study of several sensitive *SD*⁺ chromosomes reveals that the correlation also extends to the cytological phenotypes. For chromosomes with >2000 *Rsp* copies (*Rsp^ss^* chromosomes), spermatid elimination occurs at individualization, while spermatids carrying second chromosomes with fewer *Rsp* copies (≤1300; *Rsp^s^* chromosomes) tend to escape this differentiation arrest. These results support the hypothesis that a very large *Rsp* satDNA block perturbs the histone-to-protamine transition in distorter males, possibly by impeding local SNBP deposition and normal chromatin compaction. In distorter males carrying a *Rsp^ss^* chromosome, compaction defects in *SD*⁺ spermatids might reach a threshold that triggers their systematic elimination during the individualization process. Supporting a functional link between *SD* and SNBPs, knocking-down *Mst35Ba/b* or *Mst77F* induces segregation distortion in the absence of the *Sd* mutation but in the presence of the other *SD* genetic components (i.e *M(SD); E(SD)* and *St(SD)*) [31]. This observation suggests that limiting amounts of SNBPs in a sensitized genetic background exacerbate the negative impact of the *Rsp^ss^* satDNA on sperm nuclear compaction.

It is still unclear how *Rsp* satDNA perturbs the histone-to-protamine transition. Several studies suggest that spermatid differentiation defects may result from perturbation of *Rsp* satDNA transcriptional activity and/or chromatin state early during pre-meiotic stages. For instance, although the *SD* phenotype manifests post meiosis, the critical stage for the establishment of distortion is in spermatocytes [32] but see [33] and the main driver, *Sd-RanGAP*, was shown to be mislocalized in primary spermatocytes [8]. Moreover, the piRNA pathway, which is active early in spermatogenesis [34–36], may also influence *SD* through its role in regulating repeated DNA by recruiting proteins involved in establishing heterochromatin such as the H3K9 methyltransferase Eggless/SETDB1 [37–38]. Indeed, several heterozygous mutants of piRNA biogenesis pathway genes enhance distortion levels in the *SD* system [39]. In addition, the piRNA pathway regulates transcription of *Rsp* in testes and ovaries [40–41] and influences heterochromatin establishment at *Rsp* satDNA in embryos [41]. Thus, *Rsp* piRNA biogenesis may be disrupted in distorter males. For example, a mislocalized Sd-RanGAP protein may perturb the nucleo-cytoplasmic trafficking of precursor RNAs and then lead to defective *Rsp* heterochromatin establishment and/or maintenance in premeiotic cells [39,42,43]. The resulting aberrant heterochromatin organization could locally disturb histone eviction and/or SNBP incorporation in elongating spermatids. Interestingly, downregulating non-coding RNAs from the most abundant satDNA in the *D. melanogaster* genome, (AAGAG)n, in premeiotic cells, perturbs Mst77F and ProtA/B incorporation and blocks spermatid differentiation [44].

This study thus supports the notion that satDNA transcription is essential early during spermatogenesis, possibly to produce piRNA and maintain heterochromatin, to allow normal progression of spermatid nuclei through the histone-to-protamine transition. The repetitive nature of satDNA sequences may make them intrinsically difficult to pack with SNBPs and this could require a proper heterochromatin organization. Alternatively, as piRNA functions in the fly testes remain poorly understood [36], one cannot exclude the possibility that satDNA piRNAs play a direct role in the histone-to-protamine transition. In any case, what makes the specificity of *Rsp* in the *SD* system relative to other large satDNAs is still an unsolved mystery.

It has been previously proposed that spermatid individualization may represent a checkpoint to eliminate improperly differentiated spermatids [25,45,46]. Perturbation of this process has been observed in other genetic backgrounds, such as in *Mst77F* loss-of-function mutants for instance [46]. In *Gla/SD5*, *Gla/SD-Mad* and *RAL-380/SD-Mad* males, the systematic and rapid $SD^+ Rsp^{ss}$ spermatid degeneration at the time of individualization strongly implies that this is an active process. Supporting the checkpoint hypothesis, Tokuyasu and colleagues reported the presence of abnormal spermatids that fail to individualize in wildtype flies like $SD^+$ spermatids in $SD$/$SD^+$ males [27]. The checkpoint may function to selectively remove abnormal spermatids to mitigate a fecundity loss suffered by the male parent. This implies that a molecular mechanism discriminates abnormal nuclei within the cyst cytoplasm. In the *Drosophila* embryo, which starts to develop as a syncytium, damaged nuclei are eliminated during the blastoderm formation when nuclei migrate to the periphery through a Chk2-dependent checkpoint [47]. A similar mechanism might thus operate during spermiogenesis to selectively remove abnormal spermatids.

We propose that the individualization checkpoint could be activated when spermatid nuclei are not properly compacted. In our model, perturbation of the histone-to-protamine transition locally in the chromatin within the large block of *Rsp* satDNA could trigger the arrest of SNBP incorporation in the whole nucleus and its elimination (Fig 9). This model could explain why $SD^+ Rsp^s$ (with ≤ 1300 repeats) spermatids are not systematically eliminated at individualization. In this case, protamine incorporation and thus nuclear compaction would be less disturbed and might be sufficient in some nuclei to escape the checkpoint. A similar mechanism may also be involved in some interspecific hybrids where male sterility is apparently caused by defective heterochromatin state in pre-meiotic cells that result in post-meiotic defects [2,48,49] or in other meiotic drive systems [2].

*SD* is a co-adapted gene complex that involves several linked enhancers of drive [4,50–55] and both linked and unlinked suppressors that counteract drive [2,4,56–58]. While the existence and contribution of these genetic factors to levels of distortion is well documented, it seems also important to consider their impact on spermiogenesis. For instance, we have identified a weak suppressor, *Su(SD)X-380*, which substantially modifies the phenotype of spermatid elimination with a modest effect of segregation distortion levels.

Finally, beyond the interest of understanding meiotic drive systems in general, this work shows that *SD* is an excellent model to study the constraint of heterochromatin organization on the histone-to-protamine transition. Future work characterizing the chromatin organization and transcriptional activity of the *Rsp* satDNA should yield important insights into drive and spermatogenesis.

## Materials and methods

### Fly genetics

Flies were reared at 25˚C on a classical agar, yeast, corn flour fly medium. The following strains were obtained from the Bloomington Drosophila Stock Center (BDSC) and used as a source of

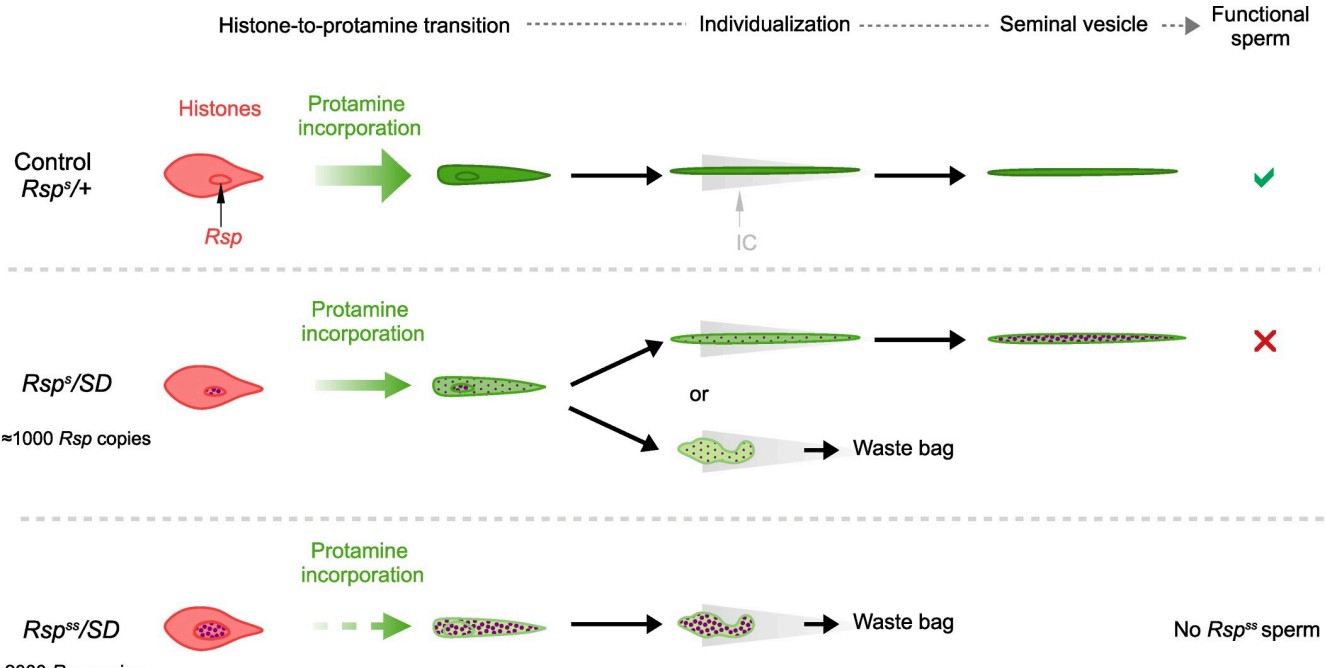

**Fig 9. A model for spermatid elimination in SD males.** In control males carrying a large *Rsp* satDNA block, *Rsp* heterochromatin organization allows the histone-to-protamine transition to occur normally. Spermatids thus individualize and are released in the seminal vesicle. In *SD* males carrying a $SD^+$ $Rsp^s$ chromosome with ca.1000 *Rsp* copies, a modified *Rsp* satDNA chromatin state (represented as purple dots) slightly perturbs SNBP incorporation. In some $SD^+$ $Rsp^s$ nuclei, chromatin compaction defects are too severe and trigger differentiation arrest. These nuclei are eliminated during individualization and end up in the waste bag. In some other $SD^+$ $Rsp^s$ nuclei, nuclear compaction defects are weaker and spermiogenesis progress normally. In this case, abnormally compacted needle-shaped $SD^+$ $Rsp^s$ sperm nuclei are released in the seminal vesicle. In *SD* males carrying a $SD^+$ $Rsp^{ss}$ chromosome ($>$ 2000 *Rsp* copies), *Rsp* satDNA chromatin defects impair SNBP incorporation. Nuclear compaction defects trigger the elimination of $SD^+$ $Rsp^{ss}$ during individualization.

*Rsp^s* chromosomes: *Gla/CyO* [*w^{1118}*; *In(2LR)Gla, wg^{Gla-1} Bc^1/CyO*; stock #5439], *iso-1* or *cn^1 bw^1* [*y^1*; *Gr22b^{iso-1} Gr22d^{iso-1} cn^1 CG33964^{iso-1} bw^1 sp^1*; *MstProx^{iso-1} GstD5^{iso-1} Rh6^1*; stock #2057], *RAL-380* (stock #25190), *RAL-313* (stock #25180), *RAL-309* (stock #28166). The two following stocks carrying the strong *SD5* and *SD-Mad* distorter chromosomes were also obtained from the BDSC: *SD5/SM1* [*In(2R)SD5, In(2R)NS, Dp(2;2)RanGAP^{SD}, RanGAP^{SD}/SM1*; stock #393] and *SD-Mad* [*SD-Mad, In(2LR)SD72, In(2R)NS, Dp(2;2)RanGAP^{Sd}, RanGAP^{Sd} E(SD)^1 Rsp^i M(SD)^1 St(SD)^1*; #64324]. The *protB-GFP* transgene was previously described [16]. To obtain *SD* males carrying *protB-GFP*, we generated the two stocks: *w^{1118}*; *SD-Mad/CyO*; *protB-GFP/TM6B* and *w^{1118}*; *SD5/CyO*; *protB-GFP/TM6B*. Males from these stocks were crossed to *w^{1118}*; *Gla/CyO* or *y*; *cn^1 bw^1* virgin females to obtain distorter males (S7A and S7B Fig).

To test distortion on the *RAL* strain second chromosomes, we first generated a marked *SD* chromosome with a *Cy* dominant marker. After meiotic recombination between *Cy Kr* and *SD-Mad* chromosomes, a strong *Cy* distorter recombinant chromosome was selected and backcrossed several times with *SD-Mad*. Since *Cy SD-Mad* is maintained in heterozygous background with *SD-Mad*, we select flies with *Cy SD-Mad* chromosome every generation (see cross scheme on S9 Fig).

## Distortion genetic tests and *k* value

To measure genetic distortion levels, single males (2–5 days old) were crossed with two virgin females and placed at 25˚C for one week before discarding the flies. For each genotype tested,

10 to 20 independent crosses were set up. In each vial, the progeny was genotyped and counted for 18 days after parents were introduced in the vial. Crosses producing less than 30 flies were not considered. The strength of segregation distortion is expressed as a *k* value, calculated as the number of flies carrying the *SD* chromosome (or the control chromosome, see Table 1) among the total progeny.

## Immunofluorescence

Testes from 2 to 5-day old males were dissected in PBS-T (1X PBS, 0.15% Triton) and fixed for 20 min in 4% formaldehyde at room temperature. For seminal receptacles, females were dissected after being placed in a vial with males in a 1:1 ratio for 2 to 3 days at 25°C. Tissues were washed three times in PBS-T and incubated with primary antibody overnight at 4°C. After three 20-min washes in PBS-T, they were incubated with secondary antibodies at room temperature for 2 hours. They were then washed three times and mounted in mounting medium (DAKO, ref #S3023) containing 1μg/mL DAPI or incubated with Phalloidin 633 (1:1000 in 1X-PBS; Phalloidin-FluoProbes 633A #FT-FP633A) for 30 min at room temperature, washed and mounted in the DAPI mounting medium. Primary antibodies used were: mouse anti-histone antibody (1:1000; Millipore ref #MABE71), rabbit Tpl94D (1:100) [46], mouse anti-dsDNA (1:3000; Abcam ref # 27156), rabbit anti-Vasa (1: 5000; a generous gift from Paul Lasko, McGill University, Canada) and secondary antibodies were a DyLight 550 conjugated goat anti-rabbit IgG (ThermoScientific; ref #84541) and DyLight 550 conjugated goat anti-mouse IgG (ThermoScientific; ref #84540). Images were acquired on an LSM800 confocal microscope (CarlZeiss) and processed using the Zen (CarlZeiss) and Fiji softwares [59]. For each experiment at least three independent crosses and immunostainings of 5–7 testis pairs were done.

## DNA-FISH

DNA-FISH of *Rsp* and *359 bp* satellites on testes from $cn^1$ $bw^1$/*SD-Mad* males (Fig 2B) were performed following [60]. Testes from 3–5 day old flies were dissected in PBS and treated for 8 min in 0.5% sodium citrate. The testes were fixed in 45% acetic acid and 2% formaldehyde for 6 min, placed on a poly-L-lysine slide, squashed, and dehydrated. Slides were denatured at 95°C for 5 min in hybridization buffer (with *Rsp* and *359 bp* satellite probes) and incubated at 37°C overnight in a humid chamber. The *Rsp* probe was a Stellaris probe from [41,61] (Rsp-Quasar570) and the *359 bp* probe was a Cy5-labeled oligo probe (5'-Cy5TTTTCCAAATTTCGGTCATCAAATAATCAT-3') previously described in [62]. Slides were mounted in SlowFade Diamond Antifade Mountant with DAPI (ThermoFisher; ref #S36964), visualized on a Leica DM5500 upright fluorescence microscope, imaged with a Hamamatsu Orca R2 CCD camera, and analyzed using Leica's LAX software.

For DNA-FISH on squashed testes (except for $cn^1$ $bw^1$/*SD-Mad* in Fig 2B) and seminal vesicles, probes were prepared as follows. A 359 bp fragment from the *359 bp* satDNA on the X chromosome was PCR amplified using the following primers 5'-CGGTCATCAAATAAT-CATTTATTTTGC-3' and 5'-CGAAATTTGGAAAAACAGACTCTGC-3' [63] and $w^{1118}$ genomic DNA as a template. The fragment was then cloned using T/A cloning in the pGEMT vector (Promega, ref#A1360). For the *Rsp* satDNA probe, we amplified the *Rsp* satDNA by PCR from *Gla/CyO* genomic DNA and cloned a ca. 700bp fragment corresponding to about four *Rsp* repeats in a pGEMT vector. Primer used were 5'-CCAGGCGAACAGAAGATACC-3' et 5'-TTTTGACCGCTTAAAATGACA-3'. *359 bp* and *Rsp* pGEMT plasmids were then used as templates to synthetize DNA probes using, respectively, the PCR labelling Cy-3 and Cy-5 kits (Jena Bioscience, ref# PP-301L-Cy3 and #APP-101-Cy5) according to the

manufacturer's instructions with M13 universal primers. Spermatozoa spreads were prepared as described in [64] with modifications. Seminal vesicles from 4 to 5-day old males were dissected in PBS, then opened with needles to spread sperm cells on home-made poly-L-lysine coated slides. Spermatozoa were then permeabilized in 1% Triton-PBS for 30 min at room temperature, treated with 2mM DTT in PBS for 30 min to reduce disulfide bonds, washed twice with PBS and incubated in fixation solution (50% acetic acid, 4% formaldehyde in PBS) for 4 min at room temperature. The slides were then dehydrated in absolute ethanol, air dried and kept at 4°C until staining. For squashed testes, tissues were dissected in PBS-0.15% Triton, incubated for 4 min in a drop of fixation solution, squashed and frozen in liquid nitrogen. Slides were then dehydrated for 10 min in absolute ethanol and air dried.

DNA-FISH was then performed as described in [65] with modifications. Slides were rehydrated in PBS for 5 min at RT, then incubated in 100mM HCl/0.1% pepsin for 90 seconds to remove proteins, washed in PBS and incubated in a 3:1ethanol/acetic acid mix for 15 min at RT. Then slides were rinsed in 2X SSC, treated with 0.2 mg/mL RNaseA in 2X SSC for 30 min at 37°C and rinsed again with 2X SSC before dehydration in successive 70%, 90% and 100% ethanol baths and air dried. Sperm cells were first incubated in a drop of 70% formamide/2X SSC and incubated at 95°C for 5 min. After denaturation, the slides were washed twice with ice cold 2X SSC for two min, dehydrated and air dried. The probes (50 ng of *Rsp* and 35 ng of *359 bp* probes per slide) were diluted in hybridization buffer (1mg/mL sonicated salmon sperm DNA (Sigma-Aldrich ref# D9156), 50% formamide, 2X SSC, 50% Dextran Sulfate). The mixture was boiled at 95°C for 5 min and quickly chilled on ice before deposition on sperm nuclei. The slides were incubated at 37°C overnight. After hybridization, the slides were washed three times in 2X SSC at 42°C for 5 min, and once in 0.2X SSC for 5 min at room temperature. They were then air dried before adding mounting medium containing 1µg/mL DAPI. Images were acquired as described for Immunofluorescence.

## Quantitative PCR

Genomic DNA from 5 males was extracted using the NucleoSpin Tissue XS kit (Macherey-Nagel, ref #740901). To amplify the *Rsp* satDNA region, we used two sets of primers: 5'-CCAGGCGAACAGAAGATACC-3' and 5'-TTTTGACCGCTTAAAATGACA-3'; and 5'-AAGTTATGTCATTTTAAGCGGTCA-3' and 5'-AACTTAGGCAATTTACTGTTTTTGC-3'. As a control, we amplified a fragment into *nup62* gene (primers 5'-GGCACCTACTGCTGGT ATCG-3' and 5'-AATCCAAAGGCTGGTGGAG-3'). Quantitative PCR analysis was performed with 5ng of template gDNA in a 25µl reaction using the TB Green Premix Ex Taq II (Takara, ref #RR820L) and the CFX Connect (Biorad CFX Connect) system. For each set of primers, standard and melting curve analyses were performed to check for, respectively, PCR efficiency and specificity. qPCR analysis was done using technical duplicates on three biological replicates. The *nup62* gene was used as internal control with a known copy number (two) so that genomic DNA levels were normalized for each sample to the levels of *nup62*. Based on [28], we considered that *cn[1] bw[1]/SD-Mad* flies carry 1000 repeats. The copy number in the *Rsp* satDNA is relative to *cn[1] bw[1]/SD-Mad* and was calculated using the comparative quantification ΔΔCT method [66].

## Supporting information

**S1 Fig. The histone-to-protamine transition in *SD5/CyO* control males.** Confocal images of individual spermatid cysts from *SD5/CyO; protB-GFP* control testes stained in (**A**) with a pan-histone antibody (red), an antibody against the Tpl94D transition protein (white) and DAPI (cyan); in (**B**) with phalloidin (F-actin; white) and DAPI (blue) and in (**C**) with an anti-

dsDNA antibody (dsDNA-Ab; red) and DAPI (blue). Scale bar: 10μm.
(TIF)

**S2 Fig. An anti-dsDNA antibody to probe for chromatin compaction.** Confocal images of a whole-mount testis stained with DAPI (blue), phalloidin (F-actin; white) and an anti-dsDNA antibody (dsDNA-Ab; red). Top panels show the nucleus of a somatic cell next to a sperm cell nucleus (arrow). While the somatic nucleus is brightly stained with the anti-dsDNA antibody, the sperm nucleus is impermeant to it. DAPI staining intensity is proportional to DNA compaction and brightly stains highly compacted DNA such as heterochromatin (yellow arrow) in somatic nuclei. On the opposite, the anti-dsDNA staining is inversely proportional to chromatin compaction and is weak in heterochromatic regions. Bottom panels show three cysts of 64 spermatid nuclei at different stages. The two cysts on top contain elongating spermatid nuclei before individualization. These nuclei are stained with the anti-dsDNA antibody. The bottom cyst contains nuclei which have been invested by individualization actin cones. At this stage, nuclei are negative for the anti-dsDNA staining because of the high compaction of chromatin. Asterisk indicates a somatic nucleus. Scale bar: 10μm.
(TIF)

**S3 Fig. Seminal vesicle contents in *Gla/CyO* control males.** (**A**) Confocal images of a whole-mount seminal vesicle from a *Gla/CyO; protB-GFP* control male stained with an anti-dsDNA antibody (dsDNA-Ab; red), DAPI (blue) and phalloidin (F-actin; white). A wide view of the seminal vesicle is shown on the left panel (scale bar: 50μm). The dashed white square corresponds to a magnified region shown on right panels (scale bar: 10μm). Almost all sperm nuclei are negative for the anti-dsDNA antibody and are thus properly compacted. (**B**) DNA-FISH performed on seminal vesicle contents with a *Rsp* probe (green) and a probe for the *359 bp* satDNA (red) on the X chromosome as a control (scale bar: 10μm). Sperm nuclei appear larger in all panels because they have been treated with DTT to facilitate probe penetration.
(TIF)

**S4 Fig. Abnormally compacted sperm nuclei from *cn¹ bw¹/SD-Mad* males are lost before storage in the female genital tract.** Confocal images of the seminal receptacle from a *w¹¹¹⁸* female mated to a *cn¹ bw¹/CyO; protB-GFP* control male (left) or a *cn¹ bw¹/SD-Mad; protB-GFP* male (right) and stained with DAPI (blue), an anti-dsDNA antibody (dsDNA-Ab; red) and phalloidin (F-actin; white). Right images show sperm nuclei stained with the anti-dsDNA antibody (arrowhead). Over ten seminal receptacles analyzed from females mated to *cn¹ bw¹/SD-Mad; protB-GFP* males, only one contained anti-dsDNA positive sperm nuclei (10/160). Scale bars: 20μm in full size images and 10μm in insets.
(TIF)

**S5 Fig. The histone-to-protamine transition in *Gla/SD-Mad* and *cn¹ bw¹/SD5* distorter males.** Confocal images of whole-mount testes from *Gla/SD-Mad; protB-GFP* and *cn¹ bw¹/SD5; protB-GFP* males stained with an anti-dsDNA antibody (dsDNA-Ab; red), DAPI (blue) and phalloidin (F-actin, white). In *Gla/SD-Mad*, many abnormally-shaped nuclei that are weakly stained with ProtB-GFP and brightly stained with the anti-dsDNA antibody are visible in pre-IC, IC and post-IC cysts. In *cn¹ bw¹/SD5* testes, abnormally-shaped anti-dsDNA positive nuclei are less frequent. Bundles of post-IC spermatid contain needle-shaped anti-dsDNA positive nuclei. Scale bar: 10μm.
(TIF)

**S6 Fig. DNA-FISH on spermatid nuclei.** (**A**) DNA-FISH on squashed testes from *cn¹ bw¹/CyO*, *Gla/CyO*, *RAL-313* and *RAL-309 and RAL-380* males performed with a *Rsp* (green) and a

*359 bp* (red) probe. Scale bar: 10μm (**B**) Box plot showing the ratio of *Rsp* signal area over nuclear area expressed as a percentage. For each genotype, the number of isolated nuclei analyzed is indicated. The area of *Rsp* signals is larger in *Gla* spermatid nuclei compared to *cn¹ bw¹* ones, in agreement with molecular quantification. Area of *Rsp* satDNA in *RAL* strains are also proportional to copy numbers determined by qPCR. Wilcoxon test, non-significant (ns) p-value>0.05, * p-value <0.05, ** p-value<0.01, ***p-value<0.001.
(TIF)

**S7 Fig. Cross schemes to obtain the different genotypes of distorter and control males.**
(TIF)

**S8 Fig. *Su(SD)X* modifies the histone-to-protamine transition and individualization phenotypes of *RAL-380/SD-Mad* males.** Confocal images of *RAL-380/SD-Mad; protB-GFP* testes carrying the X chromosome from the *RAL-380* strain with *Su(SD)X-380* suppressor (right panels) or the *w¹¹¹⁸* chromosome (left panels). Protamine incorporation and individualization appear less disturbed in presence of *Su(SD)X-380*. Bundles of individualized spermatid nuclei (post-IC) are less disturbed but include many needle-shaped nuclei positively stained with the anti-dsDNA antibody. Scale bar: 10μm.
(TIF)

**S9 Fig. Cross scheme to obtain the *Cy SD-Mad* chromosome.**
(TIF)

## Acknowledgments

We are grateful to the Bloomington Drosophila Stock Center and the Drosophila Genetic Reference Panel for fly stocks and to the SFR Biosciences (UMS3444/CNRS, US8/Inserm, ENS de Lyon, UCBL) facilities: Robert Renard and the Arthro-Tools staff for fly food preparation and the Plateau Technique d'Imagerie/Microscopie. We thank Hafida Sellou for her help in the preliminary observation of SD male phenotypes, Paul Lasko for kindly providing the anti-Vasa antibody, Marie Delattre for the Cy3 labeling kit, Guillermo Orsi and Béatrice Horard for critical reading of the manuscript and Loppin's lab for discussion and feedback.

## Author Contributions

**Conceptualization:** Benjamin Loppin, Raphaëlle Dubruille.

**Formal analysis:** Marion Herbette.

**Funding acquisition:** Amanda M. Larracuente, Raphaëlle Dubruille.

**Investigation:** Marion Herbette, Xiaolu Wei, Ching-Ho Chang, Amanda M. Larracuente, Benjamin Loppin, Raphaëlle Dubruille.

**Methodology:** Marion Herbette.

**Project administration:** Benjamin Loppin, Raphaëlle Dubruille.

**Supervision:** Amanda M. Larracuente, Benjamin Loppin, Raphaëlle Dubruille.

**Validation:** Marion Herbette, Benjamin Loppin, Raphaëlle Dubruille.

**Visualization:** Marion Herbette.

**Writing – original draft:** Raphaëlle Dubruille.

**Writing – review & editing:** Marion Herbette, Xiaolu Wei, Ching-Ho Chang, Amanda M. Larracuente, Benjamin Loppin, Raphaëlle Dubruille.

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
