## [Decision Letter · Decision Letter 0]

23 Apr 2021

Dear Dr Dubruille,

Thank you very much for submitting your Research Article entitled 'Distinct spermiogenic phenotypes underlie sperm elimination in the Segregation Distorter meiotic drive system' to PLOS Genetics.

The manuscript was fully evaluated at the editorial level and by independent peer reviewers. The reviewers appreciated the attention to an important topic and all three reviewers were enthusiastic in supporting your manuscript. However, they also identified some minor concerns that we ask you address in a revised manuscript. Specifically, please focus your attention on three aspects of the reviewers' comments: (1) Please include FISH data on control genotypes in Figure 2 (as requested by reviewers); (2) modify discussion to elaborate/clarify concept of checkpoints; and (3) please address the typos and grammatical suggestions pointed out by the reviewers.

We therefore ask you to modify the manuscript according to the review recommendations. Your revisions should address the specific points made by each reviewer.

[LINK]

Yours sincerely,

Giovanni Bosco, Ph.D.

Associate Editor

PLOS Genetics

Gregory P. Copenhaver

Editor-in-Chief

PLOS Genetics

Reviewer's Responses to Questions

**Comments to the Authors:**

Reviewer #1: I have previously reviewed this for Review Commons, and i had no major comments. This is an important paper, providing the results of foundational work to understand the fascinating phenomenon of segregation distortion, although the underlying mechanisms are yet to be discovered. PLoS Genetics would be a perfect home for this well-done paper.

Reviewer #2: In the manuscript “Distinct spermiogenesis phenotypes underlie sperm elimination in the segregation distorter meiotic drive system,” Herbette et al. show that two different strong distorting chromosome combinations have different cellular phenotypes. The Gla/SD5 combination results in sperm that are defective in the histone to protamine transition and fail to individualize properly. In contrast, the cn1 bw1/SD-Mad combination is a mixture of unindividualized sperm and those that individualize and make it to the seminal vesicle but appear defective (under-condensed) in nuclear morphology. The degree of defects appears to correspond not to the SD component but to the number of Rsp satDNA copies on the SD+ chromosome. The authors also identify a previously unknown X-linked suppressor of the SD effect. As the authors point out, the cellular basis linking severity of the SD effect with the copy number of Rsp satDNA copies is poorly understood. For this reason, the study makes a substantial contribution to this aspect. The article is well written and clear to understand. The work appears to be very well done; the genetic crosses seem to include proper control conditions and the cytological imaging is of high quality. However, there are a few questions and comments, one experimental and the others pertaining to statements made in the discussion.

The authors found improperly condensed sperm nuclei from the cn1 bw1/Sd-Mad males in the seminal vesicle. On page 15, they stated: “Although we know that these sperm cannot give rise to viable progeny, the timing of their elimination remains to be determined.” It would be interesting, important, and potentially easy to examine some embryos fertilized by strong SD males to see if these sperm are capable of penetrating the egg and if so, what happens afterward.

In the discussion, the authors state: “The checkpoint may function to selectively remove abnormal spermatids, thus avoiding the production of defective spermatozoa that could impact progeny survival.” It might be better to state that the checkpoint effect could mitigate a fecundity loss suffered by the male parent.This may be especially true if the under-condensed sperm are capable of undergoing fertilization but result in embryonic death.

How does the checkpoint work to eliminate any defective sperm if all sperm in a bundle are interconnected by a common cellular cytoplasm? It seems difficult to imagine a checkpoint that operates to eliminate individual sperm within a bundle of otherwise normally developing sperm. Is there evidence of checkpoints acting on certain individual nuclei and not others, all within a syncytium? Further elaboration on this matter would be helpful.

The authors mention the possibility that piRNA perturbation may either directly or indirectly cause defective Rsp heterochromatin formation. Why would such an effect on heterochromatin formation be specific to the Rsp satDNA? There are many other abundant satDNAs in the fruit fly genome that also presumably utilize the trans-nuclear membrane transport system and, thus, should also be similarly affected under this model.

Top of page 5 – “compaction is primary driven” should be “primarily driven”

Title on top of page 12 should read “…when Rsp carries…”

Toward the top of page 12, “…with single-molecule sequencing long reads and…” I think should read “…with single-molecule, long read sequencing and…”

Top of page 14, “…levels were lower…” Lower in comparison to what? It would help to be more specific.

Toward the top of page 14, “…stained their testes with DAPI and F-actin…” One cannot stain with F-actin, but F-actin is stained with certain reagents.

Reviewer #3: The authors have performed a careful cytological study of spermiogenesis in Segregation Distortion males, with methods and resolution beyond previous work on this well-studied system. The images are high quality and are consistent with the authors’ conclusion that partially distinct phenotypes occur in different SD genotypes. A more quantitative assessment of statements such as “a few nuclei showed …” would be ideal, but one acknowledges the difficulty of doing so for some of the assays. Because of the lack of quantification and the fact that many modifiers can affect SD penetrance and could be variable among stocks, the further conclusion that specific defects associate specifically with Rsp copy number is less firmly established (but is suggestively supported). The manuscript is well written and easy to understand.

Previous Rev 1 had some good points but it doesn’t appear that all suggestions were implemented by the authors. For example, the authors say they were going to modify Fig 2 to include FISH data on control genotypes, but this doesn’t appear to have been done.

In Fig 1, it looks like more than a few nuclei staining with Rsp are normal in shape.

Minor points.

p.6 ‘classical’ is unnecessary, what would a non-classical chromosome be?

p.7 should be ‘and a few nuclei with an apparent …’

p. 9 stained-somatic should be stained somatic

p. 12 section title at top. Should be Rsp carries 2000…

p. 16 and then lead to …

Fig 1 legend. Spermatids become individualized and coiled …; appear abnormally shaped and are eliminated …

Fig 2 legend. Cn bw / Sd-Mad should be described before Gla/SD5 in order to match left-to-right order of the figure.

Fig 3 legend. Is variable from one cyst to another …

Fig 8 legend. “too important” should be “too severe” or similar. Also, what does the purple coloring indicate in the two SD genotypes.

**Have all data underlying the figures and results presented in the manuscript been provided?**

Reviewer #1: Yes

Reviewer #2: Yes

Reviewer #3: Yes

PLOS authors have the option to publish the peer review history of their article (what does this mean?). If published, this will include your full peer review and any attached files.

Reviewer #1: **Yes: **Yukiko Yamashita

Reviewer #2: No

Reviewer #3: No

---

## [Editor Report · Decision Letter 1]

10 Jun 2021

Dear Dr Dubruille,

We are pleased to inform you that your manuscript entitled "Distinct spermiogenic phenotypes underlie sperm elimination in the Segregation Distorter meiotic drive system" has been editorially accepted for publication in PLOS Genetics. Congratulations!

Yours sincerely,

Giovanni Bosco, Ph.D.

Associate Editor

PLOS Genetics

Gregory P. Copenhaver

Editor-in-Chief

PLOS Genetics

Comments from the reviewers (if applicable):

**Data Deposition**

http://datadryad.org/submit?journalID=pgenetics&manu=PGENETICS-D-21-00398R1

**Press Queries**

---

## [Editor Report · Acceptance letter]

29 Jun 2021

PGENETICS-D-21-00398R1 

Distinct spermiogenic phenotypes underlie sperm elimination in the *Segregation Distorter* meiotic drive system 

Dear Dr Dubruille, 

We are pleased to inform you that your manuscript entitled "Distinct spermiogenic phenotypes underlie sperm elimination in the *Segregation Distorter* meiotic drive system" has been formally accepted for publication in PLOS Genetics! Your manuscript is now with our production department and you will be notified of the publication date in due course.

With kind regards,

Andrea Szabo

PLOS Genetics

On behalf of:
